# Four-Types of IIT-Induced Group Integrity of *Plecoglossus altivelis*

**DOI:** 10.3390/e22070726

**Published:** 2020-06-30

**Authors:** Takayuki Niizato, Kotaro Sakamoto, Yoh-ichi Mototake, Hisashi Murakami, Takenori Tomaru, Tomotaro Hoshika, Toshiki Fukushima

**Affiliations:** 1Faculty of Engineering, Information and Systems University of Tsukuba, Tennodai 1-1-1, Tsukuba, Ibaraki 305-8573, Japan; tomo.tkb1@gmail.com (T.H.); toshiki_fukushima@outlook.jp (T.F.); 2Leading Graduate School Doctoral Program in Human Biology, University of Tsukuba, Tennodai 1-1-1, Tsukuba, Ibaraki 305-8573, Japan; 3The Institute of Statistical Mathematics, Tachikawa, Tokyo 190-0014, Japan; motokake@mns.k.u.tokyo.ac.jp; 4Research Center for Advanced Science and Technology, University of Tokyo, Tokyo 153-0041, Japan; hssh415@gmail.com; 5Department of Computer Science and Engineering, Toyohashi University of Technology, Aichi 441-8580, Japan; tomaru.take@gmail.com

**Keywords:** integrated information theory, collective behaviour, self-organization, cause and effect structure

## Abstract

Integrated information theory (IIT) was initially proposed to describe human consciousness in terms of intrinsic-causal brain network structures. Particularly, IIT 3.0 targets the system’s cause–effect structure from spatio-temporal grain and reveals the system’s irreducibility. In a previous study, we tried to apply IIT 3.0 to an actual collective behaviour in *Plecoglossus altivelis*. We found that IIT 3.0 exhibits qualitative discontinuity between three and four schools of fish in terms of Φ value distributions. Other measures did not show similar characteristics. In this study, we followed up on our previous findings and introduced two new factors. First, we defined the global parameter settings to determine a different kind of group integrity. Second, we set several timescales (from Δt=5/120 to Δt=120/120 s). The results showed that we succeeded in classifying fish schools according to their group sizes and the degree of group integrity around the reaction time scale of the fish, despite the small group sizes. Compared with the short time scale, the interaction heterogeneity observed in the long time scale seems to diminish. Finally, we discuss one of the longstanding paradoxes in collective behaviour, known as the heap paradox, for which two tentative answers could be provided through our IIT 3.0 analysis.

## 1. Introduction

Global dynamic patterns in swarming [1,2,3,4,5,6], schooling (of fish) [7,8,9,10,11], flocking (of birds) [12,13,14,15,16,17], genes [18], proteins [19], or neural networks [20,21,22,23], emerge from local interactions between self-organising individuals or components. Despite their complexity, these collective systems are capable of processing information efficiently in critical conditions; for example, individuals making a swift response [14,15,16] or a group making a good decision [24,25,26] in a changing environment. Individual conflicts do not necessarily disrupt group integrity, but contribute to their collective response [27,28,29]. The complexity is characterised using statistical models and information theory. Self-organised criticality (SOC) [30] is considered a convenient strategy that is evolutionarily adopted by living systems to efficiently cope with their external environment [31,32,33,34,35]. However, for smaller system sizes, heterogeneous interactions [36,37] result in different global motion patterns in N=2 and N=3 [38,39]. Hence, some studies thoroughly investigated information transfer (or causal relationships) among individuals in small groups [40,41,42,43,44,45,46,47,48,49,50].

Most of the previous studies on dynamical systems including collective behaviours focus on ‘what the system does’ and the system’s dynamical trajectory through its state space; that is, they consider an extrinsic perspective on what is observed. Albantakis and Tononi shifted their focus to a system’s intrinsic/causal structure and considered ‘what the system is’ from its own intrinsic/causal perspective [51] based on the integrated information theory (IIT) [52,53]; they demonstrated that the intrinsic/causal complexity (as quantified by integrated information Φ) correlates well with dynamical/statistical complexity. The integrated information, represented by Φ, is defined as a degree of information loss (or an increase in uncertainty [54]) caused by the removal of certain edges of the Bayesian network [55,56,57,58,59,60,61,62]. The metric was successful in not only distinguishing various states of lost consciousness, such as dreamless sleep [63], general anaesthesia [64], or vegetative states [65] but also the different states of complex systems, such as the generalised Ising model [66], coupled oscillators [67], and coupled mapping [68].

The aim of IIT 3.0, as we mentioned earlier, is to try to shift the paradigm from ‘what the system does’ to ‘what the system is’ [51,60,61,69,70,71]. The former tries to analyse the system on the basis of its external behaviour, whereas the latter does the same according to its intrinsic causal structure. This difference is covered by the analyses of cellular automata. The cellular automaton refers to the time-development system defined using binary states while applying simple rules. Despite its simple construction, it has rich connotations. In general, a cell’s behaviour can be classified into four classes: Class I (uniform stable state), Class II (periodic or non-uniform stable state), Class III (chaos), and Class IV (complex system as the universal computation or edge of the chaos) [72,73]. In the past decades, nearly all studies on the cellular automaton focused on their behavioural differences [45,74,75], although some exceptions exist [76]. The IIT 3.0 application of Albantakis et al. [51] succeeded in classifying these four patterns without referring to their external cell behaviour (‘what the system does’). Their analyses were based on a low number of cells (up to six cells), which is insufficient to show the behavioural difference among the rules. Therefore, there is no room for the external behavioural analyses here. Furthermore, Albantakis et al. also indicated the possibility of the existence of an unobserved new class from the intrinsic causal cell structure (‘what the system is’).

The application of IIT 3.0 to a real system has at least two important implications. First, applying IIT 3.0 enables us to quantify the movement of the fish schools using its internal cause–effect structure and without reference to their external behaviour. Second, the real data analysis can connect its internal cause–effect structure to its actual external behaviour. This was partially achieved in our previous study [77], in which we applied IIT 3.0 to schools of two to five fish (*Plecoglossus altivelis*) and found that 〈Φ(N)〉 distribution was discontinuous between three and four schools of fish in terms of the emergence of leadership.

In other words, the discontinuity between three and four schools of fish in IIT 3.0 was never observed on the mutual information and the sum of the transfer entropy. We applied the same analysis to the Boid model in certain parameter settings. The result obtained using IIT 3.0 was never replicated (no discontinuity between three and four fish schools observed). Furthermore, a randomly generated network (homogeneous Markov models) never grew the standard variance similar to that of real fish schools; however, both the average Φ values increased (we use this result for our analysis in this paper).

One of the most critical points in our results is that the IIT-induced leadership must be strictly distinguished from the conventional leadership, such as that defined by the transfer entropy. First, the IIT-induced leadership refers not to group behaviour, as in other methods, but the degree of autonomy (i.e., group integrity). As we have mentioned above, IIT methods do not refer to the individual external behaviour but to its internal cause–effect structure. Second, in the conventional leadership relation, the notion of the leader and its follower is paired (i.e., one-to-one); however, the IIT leadership includes the leader and the rest of the members (i.e., one-to-many).

In this study, we extended our previous approach to examine how integrated information changes under different timescales by adding new parameter settings as the global parameter settings. By applying these metrics, we succeeded in classifying the different sizes of schools as qualitatively different autonomous systems (e.g., chasing, fission–fusion, leadership, and interactive). We also succeeded in defining these classifications based on the system’s group integrity while using minimal information partition (see Section 2.1), and related our class to the actual behaviours. This is a novel type of classification of collective animal behaviour. Finally, we discussed the heap paradox on collective behaviour, for which we showed that our IIT 3.0 analysis can provide two kinds of tentative answers.

## 2. Results

### 2.1. Brief Summary of IIT 3.0

First, we provide a brief summary of IIT 3.0. Let be the state vector, $X_{t}$, comprising binary variables $x_{t_{i}}$, where i=1,2,…,n(n∈N) at time *t*. The IIT models a system, *S*, using the discrete time multivariate stochastic process comprising *N* interacting components as follows:(1)$$p(X_{0},X_{\Delta t},\ldots ,X_{t},X_{t+\Delta t},\ldots ,X_{T}),$$
which satisfies the Markov property,
(2)$$p\left(X_{0},X_{\Delta t},\ldots ,X_{t},X_{t+\Delta t},\ldots ,X_{T}\right)=p\left(X_{0}\right)\prod_{t=\Delta t}^{T}p\left(X_{t}|X_{t-\Delta t}\right).$$

Such a discrete Markovian dynamical system, *S*, is defined using a directed graph of interconnected nodes (in this study, we assumed a complete graph) and its transition probability matrix (TPM), which specifies the conditional probability distribution, $p(X_{t}|X_{t-\Delta t})$.

A joint distribution, $p_{\mathrm{cause}--\mathrm{effect}}$, is defined as
(3)$$p_{\mathrm{cause}--\mathrm{effect}}\left(X_{t-\Delta t},X_{t}\right):=p_{u}\left(X_{t-\Delta t}\right)p_{\mathrm{effect}}\left(X_{t}|X_{t-\Delta t}\right).$$

The marginal distribution, $p_{u}\left(X_{t-\Delta t}\right)$, is a uniform distribution, which provides the maximum entropy distribution.

From the aforementioned joint probability, the backward transitional probability distribution can be defined as follows:(4)$$p_{\mathrm{effect}}\left(X_{t-\Delta t}|X_{t}\right):=\frac{p_{\mathrm{cause}--\mathrm{effect}}\left(X_{t-\Delta t},X_{t}\right)}{\sum_{X_{t-\Delta t}}p_{\mathrm{cause}--\mathrm{effect}}\left(X_{t-\Delta t},X_{t}\right)},$$
and the forward transitional probability distribution is defined as follows:(5)$$p_{\mathrm{cause}}\left(X_{t}|X_{t-\Delta t}\right):=p\left(X_{t}|X_{t-\Delta t}\right).$$

These are constructed and referred to as the cause repertoire and the effect repertoire of state $X_{t}$, respectively, which are calculated for a set of nodes within the subsystem, for mechanism M⊆S over another set of nodes within the subsystem, or for a purview of the mechanism.

After evaluating the information of the mechanism over a purview, we consider the integrated information, $\varphi_{\mathrm{cause}--\mathrm{effect}}$, of a set of system elements at state *X*. The integrated information can be defined as follows:(6)$$\phi_{\mathrm{cause}--\mathrm{effect}}:=\min\left(\phi_{\mathrm{effect}},\phi_{\mathrm{cause}}\right)$$
(7)$$\phi_{\mathrm{effect}}:=\min_{i\in I}D\left(p_{\mathrm{effect}}||p_{\mathrm{effect}}^{\left(i\right)}\right)$$
(8)$$\phi_{\mathrm{cause}}:=\min_{i\in I}D\left(p_{\mathrm{cause}}||p_{\mathrm{cause}}^{\left(i\right)}\right),$$
where the system is decomposed in all possible ways into *I*. The *D* is a measure of comparing one probability distribution to another distribution. In III 3.0, an earth mover’s distance is used as a measure.

Integrated information φ is estimated by quantifying the extent to which the cause and effect repertoires of the mechanism–purview pair can be reduced to the repertoires of its components. The amount of irreducibility of a mechanism over a purview with respect to a partition is quantified as the divergence between an unpartitioned repertoire *p* and a partitioned repertoire $p^{\left(i\right)}$. The partition that yields the minimum irreducibility is known as the minimum-information partition (MIP). Integrated information φ of a mechanism–purview pair is defined as the divergence between the unpartitioned repertoire and repertoires partitioned by the MIP. The maximum φ value is then determined for overall possible purviews to determine the maximally irreducible cause and maximally irreducible effect specified by a mechanism.
(9)$$\varphi_{\mathrm{cause}}^{\max}:=\max_{j\in C}\left(\varphi_{\mathrm{cause}}^{j}\right),\varphi_{\mathrm{effect}}^{\max}:=\max_{j\in C}\left(\varphi_{\mathrm{effect}}^{j}\right),$$
where $C=2^{N}-1$ (in this study, we adopted the ‘cut one’ approximation that only evaluates 2N bipartitions, thereby severing the edges of a node to the remaining network).

The φ value of the entire concept or the maximally integrated cause–effect information is the minimum of the maximally integrated cause information, $\varphi_{\mathrm{cause}}$, and maximally integrated effect information, $\varphi_{\mathrm{effect}}$.
(10)$$\varphi_{\mathrm{cause}--\mathrm{effect}}^{\max}:=\min\left(\varphi_{\mathrm{cause}}^{\max},\;\varphi_{\mathrm{effect}}^{\max}\right)$$

If the mechanism’s maximally irreducible cause has $\varphi_{\mathrm{cause}}>0$ and its maximally irreducible effect has $\varphi_{\mathrm{effect}}>0$ (equivalently, $\varphi_{\mathrm{cause}--\mathrm{effect}}^{\max}>0$), the mechanism is said to specify a concept.

A maximally irreducible cause–effect structure comprising concepts is known as a maximally irreducible conceptual structure (MICS), or conceptual structure (constellation of concepts) C(S) (the set of all concepts specified by the subsystem characterising all the causal constraints intrinsic to the physical system) by simply iterating the computation of the concepts over all the mechanisms, M∈P(S), where P(S) is the power set of the subsystem nodes.

Integrated conceptual information, Φ (also known as big phi, a measure of the system’s strong/integration irreducibility), is estimated by partitioning the set of elements into subsets with unidirectional cuts. The unidirectional bipartitions, $P_{\to }=\left\{S^{\left(1\right)};S^{\left(2\right)}\right\}$, of the physical system *S* are obtained by partitioning the subsystem into two parts, namely, $S^{\left(1\right)}$ and $S^{\left(2\right)}$ and cutting the edges from these parts (the connections are substituted with noise). We then calculate the conceptual structure of the partitioned system, $C(S^{P_{\to }})$, and compare it with C(S) to evaluate the difference caused by the partition. An MIP (a search over all possible directed partitions) is then performed to identify the partition that causes the least amount of difference to the conceptual structure. Φ measures the irreducibility of a conceptual structure by quantifying the difference created by the MIP to the concepts and their φ values of the system.
(11)$$\Phi =\min_{P_{\to }}D\left(C\left(S\right),C\left(S^{P_{\to }}\right)\right)$$

The difference *D* between the two conceptual structures was evaluated by an extended version of the earth mover’s distance, which is the cost of transforming a conceptual structure, C(S), into another $C(S^{P_{\to }})$ in a concept space. The candidate subset with the greatest Φ value is known as the main complex of the system *S*.

We give the definitions of the IIT 3.0 terms used for the analysis in this paper (integrated information Φ and minimum information partition) and how they are applied to the real fish schools.

MIP: The method of partitioning the system into two disconnected halves such that there is a minimal degree of information loss in the causal power of the system (the partition is unidirectional; thus, it only cuts the connections from one half to the other, but not the other way around). The cut is not bidirectional, but unidirectional.Integrated information Φ: A degree of information loss in the causal power of the system when an MIP cut is applied. High Φ values represent a strong integrated system. The reason for this is no matter how the system is divided, information loss always exceeds the value of the MIP being applied.

As Oizumi et al. proposed [57], the high Φ values also indicate the feed-back information structure of the system. The reverse information flow is automatically greater than the MIP cut flow because the MIP cut is unidirectional. Meanwhile, the low Φ values indicate the forward information structure of the system. This indicates that the greater reverse information flow exists compared with the MIP cut flow (it is also possible that the flow in both directions is equally weak).

In this paper, we consider these Φ values as an expression of the fish school’s “group integrity”. We have two reasons for this interpretation. First, Φ measures the minimum information loss by the MIP cut. In real fish schools, the information loss by the MIP cut directly corresponds to the integrity of the group behaviour because the MIP divides the fish school into two sub-schools. Second, as we confirmed earlier, the Φ values also connected with the feed-back and feed-forward structure of the system. This systematic analysis enables us to examine the internal causal structure through the relation between the MIP cut and Φ values. The latter aspect gives us an important key to analyse the relation between the actual group behaviour and IIT 3.0 [77]. For example, the position of the MIP cut in the fish school corresponds to the leadership and the rest of its members in the case where the group size is greater than four [77]. In the previous study, we showed that when the group forms the leadership structure, we confirmed the feed-forward structure (low Φ values) in the system. Therefore, the MIP cut is a useful tool to analyse the actual group behaviour.

### 2.2. Definition of the Local Parameter Settings

We tracked the trajectories of *Plecoglossus altivelis* fish schools of N = 2, N = 4, and N = 5 with three samples each, and N = 3 with four samples (10–15 min recording length; see in Section 4.4). The cohesion rate defined in Table 3 indicates the group data used in this analysis are sufficiently dense to form a stable group. To apply IIT 3.0 to schools (of fish), it is necessary to first determine their binary states because they do not possess a specific firing state such as neurons in the brain. In our previous study, we proposed the definitions of binary states for each parameter as a threshold for ON states [77]. For example, the biological meaning of such discretisation corresponds to the ON/OFF state of the individual’s contact. This individual contact is important in the system because mutual contact promotes efficient group communication, as Murakami et al. found in their study of a group of *Plecoglossus altivelis* [10]. Therefore, it is reasonable to assume that mutual contact within a flock reflects a biological meaning to the flock.

In this study, we provide a summary of the two types of parameter settings (for the detailed mathematical definition, refer to the Materials and Methods). First, the set of local parameters comprises three types of interactions: a distance (i.e., contacting other individuals), field view (i.e., observing other individuals using one’s sight), and turning rate (i.e., state of being affected by other individuals’ movements). These three parameters reflect the local interactions, such as Boids, which are often observed in various flocking models. Second, the set of global parameters comprises two types of interactions: the distance from the centre of mass and difference from the averaged group direction (explained in detail in Section 2.3). Here, we focus on only the local parameter settings.

After obtaining a binary state of those parameter settings for each individual, we apply a conjunction (or the AND operator) to the three bits of information as collective states at time *t*. For example, if the states of distance, visual field, and turning rate are ON, OFF, and OFF, respectively, for fish *i*, then *i* is in the OFF state. The application of the same procedure to all the *N*-fish in the group gives a collective state at time *t*. The central interest of this study is not an individual’s perceptual abilities (e.g., the fixed visual field), but the individual’s effective perceptions under the dynamical interactions as parameter settings. More precisely, the interaction radius and visual field should be interpreted as the ’effective’ interaction radius and the ’effective’ visual field, respectively [13].

The application of the conjunction has critical biological implications because an analogy from the Boids model, referred to by several researchers, was adopted in a collective animal behaviour as collective states [32,78,79,80,81,82]. Nearly every interaction in the general Boids model adopts the interaction radius with blind spots, then the agent’s interaction with its neighbours within its radius determines its turning direction. The conjunction of the above-mentioned three parameters replicates this statement; it would lose meaning if we replace the AND operator with the OR operator. For instance, when the two parameters (distance and visual field) are combined with AND, and the remaining (turning rate) is combined with OR (i.e., OR (AND (Distance, Visual Field), Turning parameter)), the system allows the ON state irrespective of the states of either of the two parameters (distance or visual field). This example violates the notion of the interaction distance with a blind spot.

The timescale has several implications in our analysis. In the timescale, the position vectors for every Δt step of each fish are considered (Figure 1). The longer the timescale, the more predictable and regular the trajectories in this setting are. In our previous study, we restricted our primary focus to a very short timescale (Δt=5/120 s) and a more narrowly ranged parameter setting (for the distance and visual field parameters only). A larger amount of parameter settings reveals more precise and minute causal structures, as shown later in the text. The TPM, which is assumed to have conditional independence [57,62], is obtained from the entire sequence of the collective states (Figure 2). The PyPhi [70] computes all the IIT related values using this TPM. These bit sequences as collective states are not only sensitive to parameter selection but also to timescale selection, i.e., the length of the time step. In this study, we examined an extensive range of timescales, such as Δt=5/120, 10/120, 20/120, 40/120, 80/120, and 120/120 s (see Table 3 in the Materials and Methods).

In addition to the TPM, the network structure of the group must also compute IIT-related values. Therefore, we set the network structure of the school (of fish) similar to all the connected networks without self-loops. This is because the minimum distance between any pair of fish is less than 5 mm in our experiment (see Table 3). It would be sufficient to assume that some type of interaction occurs amongst all the fish by mutual contact throughout a series of an event (i.e., approximately 10–15 min of free swimming). We chose this network structure in this study because according to the results of our previous study, for the network structure including a self-loop, the MIP cut showed weak correspondence to the leadership in the group compared with a non-self-loop network [77]. The self-loop, when included in a network, reflects the past history of each movement (e.g., direction changing). Considering that the past states to the network may fail to determine the pure inter-relations (only depends on other agent’s states) amongst fish in the group, the fully connected structure without a self-loop is the most suitable structure for the IIT 3.0 analysis.

At this point, we comment on the ambiguity caused by binarisation of states. IIT 3.0 draws intrinsic causal relationships from the network structure. In this study, we assume binary states of individuals: ON states for pairs of individuals in a Boids-like interaction and OFF states for pairs of individuals not in such an interaction. Therefore, we do not distinguish whether an OFF state is due to “not changing one’s direction” or “no other individuals in one’s visual field”. ON and OFF states inside the system are thoroughly determined by the state transitions of the whole network and their representations are interpreted by the relationship with external behaviours. As discussed in later sections, the ambiguity observed among ON and OFF states reveals abundant causal structures not observed in previous studies. Ambiguity that such binarised states inherit is adopted unintentionally or intentionally in other studies. In a similar work that used IIT 3.0 to analyse a biological network (boolean network) [70], for an OFF state individual, one cannot determine whether it is an effect of inhibition or no inputs. Each element or subset has a role determined by the activity of the whole network. As an intentional case, Gunji et al. indicated that it is unavoidable to discuss whether ON and OFF states of cellular automata is fundamentally a figure-ground relationship or simply a bit-sequence when analysing Class-4 behaviour [76]. Even though spikes of neurons are binary, we have to examine the relationship with external behaviours. The ambiguity caused by binarisation remains an avoidable problem. Connecting binary states and external behaviour is not one-to-one but one-to-many [60,61,63,64,65]. Hence, collective behaviour where external behaviour and binary state transition have a relatively direct connection is a good model to examine the range and validity of IIT 3.0.

Each collective state has a unique corresponding computed Φ value in the above-mentioned procedure. For example, in the case of a two-fish school, in which all collective states are (0, 0), (1, 0), (0, 1), and (1, 1), we have four values of Φ, that is, $\Phi_{00},\Phi_{10},\Phi_{01}$, and $\Phi_{11}$, respectively. In this study, we mainly focused on the average Φ values (denoted as 〈Φ(N)〉) for all collective states ($2^{N}$ for *N* fish schools). We also used $\sigma {\left(\Phi \left(N\right)\right)}^{2}$ as the variance of all $2^{N}$Φ values.

#### 2.2.1. Φ Values for Local Parameter Settings Across Timescales

In this study, we focus on the parameters that maximise the average value of Φ. We have three motivations. First, there are several precedents that analyse the parameters (e.g., time-delays) that maximise the transfer entropy or the integrated information [42,83]. Second, the aim of this study is to understand the intrinsic dynamics in a living system. Many high Φ values imply that the system has abundant and diverse feedback structures. Inversely, low values of Φ or zero Φs imply independent or feedforward processes. Numerous studies have indicated that information feedback structures produce diverse behaviours of biological systems [84,85]. Therefore, it is natural to focus on parameter settings that can achieve high Φs on an average to capture the strong feedback structures. Third, many recent studies have indicated a relationship between integrated information and critical phenomena [51,66,86,87,88]. In addition, the fish school of *Plecoglossus altiveliss* are known to be in a critical state [35]. Accordingly, we have a strong motivation to elucidate the type of information exchange between individual using parameters with high Φ values.

Figure 3 shows an example of 〈Φ(N)〉 distributions for the local parameter settings when the timescale is Δt=20/120 s. The ball size and gradation of the blue colour represent the 〈Φ(N)〉 values for each cell. We omitted the low 〈Φ(N)〉 values under specific values (Figure 3) from the graph for their visibility. The dense location of 〈Φ(N)〉 is important for our analysis because the 〈Φ(N)〉 values represent the group’s integrity in the school for the above-mentioned reasons (see the Section 2.1). The more the system of interest obtains the feed-back information structure (i.e., mutually connected with each other), the more that system has high 〈Φ(N)〉 values. In contrast, the areas with low 〈Φ(N)〉 values correspond to few feed-back structures (only feed-forward or mutual independent) as an autonomous system.

Before detailing our analysis, let us take note of some of the remarkable features that can be observed in Figure 3. First, the distribution of a school of two fish differs from other distributions. High 〈Φ(N)〉 values are distributed on a small visual-field area ($\xi_{VF}<\pi /2$) in a two-fish school. In three- to five-fish schools, the 〈Φ(N)〉 distribution tends to be distributed near a large $\xi_{VF}$ area. In our previous study, under particular conditions, especially in the region of $\pi <\xi_{VF}<2\pi )$, we confirmed the emergence of a leadership relationship. A high $\xi_{TR}$ value did not contribute to group integrity for any of the group sizes. In high $\xi_{TR}$ regions of more than 0.05 rad/step, no high 〈Φ(N)〉 values were confirmed. This tendency has two implications: signalling only the intense direction changes (high $\xi_{TR}$ value) is not helpful for group integration, and the ON/OFF states are not symmetric owing to the combination of other parameters. If the relation is symmetrical, 〈Φ(N)〉 values of the lower and higher $\xi_{TR}$ regions would show the same distribution; Such symmetries could not be confirmed. The interpretation of ON/OFF could differ according to the given conditions. The latter will be discussed in ‘*chasing*’ relation to a school of two fish.

#### 2.2.2. High Average Group Integrity 〈Φ(N)〉 in the Four- and Five-Fish Schools When They Have Blind Spots

As the examination and comparison of all the scales and parameter settings are difficult, we divided some regions in the parameter space for convenience. First, we divided the regions with $\xi_{TR}<\;0.05$ and $\xi_{TR}\geq 0.05$ because the 〈Φ(N)〉 values are very low in the high $\xi_{TR}$ regions (≥0.05). The difference in the 〈Φ(N)〉 distribution is mainly distributed on the low $\xi_{TR}$ regions (Figure 3; the overall results are as shown in Appendix A.)

Figure 4a presents one of the main essence in the local parameter settings. Each point in Figure 4a represents the weighted average position at which the *x*-axis represents the distance ($\xi_{D}$) and the *y*-axis represents the visual field ($\xi_{VF}$). We can visually divide the distributions into two groups: N=2,3 and N=4,5. We refer to this discontinuity in the 〈Φ(N)〉 distributions between three- and four-fish schools as ‘the emergence of leadership’ because the mean 〈Φ(N)〉 values with blind spots (i.e., $\xi_{VF}\neq 2\pi$) in the school are high only when the group size is greater than four (Figure 4 (ii); the statistical tests presented in Appendix A: only two- and three-fish schools show no statistical difference).

In our previous study, we have defined ‘leadership’ in the group satisfying the following two properties: (i) the leader is the head position of the group (i.e., the positional leader) and (ii) there is asymmetrical information flow between the leader and the rest of the members. In fact, our previous analysis found that the head along the average group direction (leadership property (i)) is strongly correlated with MIP cuts, especially in a fully connected network without self-loops [77]. This MIP cut position indicates a feed-forward relationship between the leader and its followers because the MIP cut is performed unidirectionally, especially when the corresponding Φ value is low (in the pyphi expression as {1}⇏{2,3,4,5} or {2,3,4,5}⇏{1} when the leader index is 1); this is selected as the weakest link in the network structure. In other words, the opposite information flows are always larger than the first cut (leadership property (ii)).

In Section 2.5, we will apply this method to all group sizes fish schools. Here, we just point out that the average group integrities become high only when the group size exceeds four and indicate that this result will link to the group leader.

#### 2.2.3. High Average Group Integrity 〈Φ(N)〉 in The Two-Fish Schools When Their Visual Fields Set Narrow

Next, we focus on the $0<\xi_{VF}<\pi$ regions. In these regions, only the schools of two fish show distributions for high 〈Φ(N)〉 values. The parameter distributions in Figure 5a show a relative contrast with the same distributions depicted in Figure 4a. Only two-fish schools show high 〈Φ(N)〉 values over 0.4*${\langle \Phi \left(N\right)\rangle }_{MAX}$; in contrast, no high 〈Φ(N)〉 values were observed in over four-fish schools.

This visual consideration can be confirmed in Figure 5b. We normalised all the 〈Φ(N)〉 values using ${\langle \Phi \left(N\right)\rangle }_{MAX}$ for each group size. The mean normalised Φ value of N=2 (the mean 〈Φ(N=2)〉≃0.29) in this region (Figure 5b) is as high as that in N=5 (the mean 〈Φ(N=5)〉≃0.27; Figure 4b). We found that only the mean 〈Φ(N)〉 in $\xi_{VF}$ lower regions showed significant difference to those of the remaining group (see Appendix A). Therefore, the high integrity of this lower $\xi_{VF}$ in the region of two-fish schools indicates the existence of important properties in terms of group integrity.

We refer to this tendency, which is only observed in the school of two fish, as “chasing” owing to the following reasons. First, the narrow regions of $\xi_{VF}$ parameters correspond to high 〈Φ(N)〉 values in the school of two fish. Compared with the leadership in four- and five-fish schools, the narrower $\xi_{VF}$ value in a two-fish school shows the tracking of the leader to avoid losing its sight. This leader is always positioned ahead of the follower by definition. Second, the MIP cut point also functions as an asymmetric information processing between the leader and its follower (in the pyphi expression as {1}⇏{2} or {2}⇏{1} when the leader index is 1). As the MIP cut in the school of two fish functions exclusively to each other, the asymmetric information flow always exists between the two fish.

### 2.3. Φ Values for Global Parameter Settings Across Timescales

In this section, we examine the group integrity of two global parameters (Figure 6): the angle difference ($\Xi_{AD}$) between the direction of each fish and the average group direction at time *t*, and the distance ($\Xi_{CM}$) between the position of a fish and the centre of mass of the group at time *t*. These two parameters ($\Xi_{AD}$ and $\Xi_{CM}$) determine the collective states in this condition. The collective state at time *t* in the global parameter settings is determined after calculating the conjunction (AND operator) of the two parameters as in the local parameter settings. Again, we compute TPM from a series of collective states with a fully connected network (no self-loop included) and obtain the corresponding Φ value to each collective state for the global parameters.

The two aforementioned averaged values are related to the measure of the collective behaviour domain. The average group direction is often used as the group polarity (i.e., the degree of alignment) [4,13,15,29], whereas the centre of mass is a useful measure for group clustering [6,14,89,90,91]. These interactions result in keeping a certain distance (direction) with the global state. For example, if a fish’s distance from the centre of mass is always above its threshold ($\Xi_{CM}$), then the fish cannot keep contact with the belonging group. If a fish’s direction nearly always differs from the average group direction ($\Xi_{AD}$), then the fish again gradually diverts from the school to which it belongs. This is why the use of the AND operator makes sense for the collective state in the global parameter settings. If we change the AND connective to OR, then we cannot ensure this group’s coherence.

There is a sharp difference between the local and global parameter settings in this concept. To refer to the average values, a priori is needed to assume that the group members have already been determined in advance. The average values for comparison contain the information of all the members at time *t* despite their apparent pairwise interactions (because ON/OFF state is determined using the individual fish state and its global reference). In terms of a complex system, this global parameter setting works as feed-back information from the entire school to each fish.

We can interpret the difference between the local and global parameters as follows. First, the collective state of the global settings works as a top-down system because the global reference can play the role of a feed-back system to each member. As such, the group integrity is determined by the degree of connections with the central information (i.e., averaged values) through the continuous adjustments with the global reference. In contrast, the collective state of the local parameter settings works in a bottom-up approach because there is no assumption of the group in advance. Instead, the collective state is determined as a building block through local interaction (this concept matches with the concept of Boid interaction). Both parameter settings represent typical ways of information processing (i.e., bottom-up and top-down), extensively observed in the collective behaviour [2,9,13,17].

Figure 7 shows some similarities and dissimilarities in the graph of the local parameter settings. In terms of the 〈Φ(N)〉 values, both Figure 3 and Figure 7 show an increase in 〈Φ(N)〉 according to the group size; however, in terms of the relation of inter-distributions of heat maps, the discontinuity observed in Figure 3 is not seen in Figure 7 (Appendix A). Instead, the discontinuity of 〈Φ(N)〉 distributions in Figure 7 appears between two- and three-fish schools (Appendix A: the matrix distance).

These observations imply that both the degrees of the group integrities in a school as well as the discontinuity in the 〈Φ(N)〉 distribution highly depend on the choice of the parameter settings. The local and global parameter settings indicate that different kinds of the group integrity emerge through their different interaction. The first is the interaction between each fish (the local parameter settings), and the second is the interaction at the global reference assuming that each fish in the group knows all the group members a priori (the global parameter settings). These contrasts in the interactions lead to differences in the integrity of their systems.

Next, we examine the relationship between 〈Φ(N)〉 generated by the global and the local interactions (henceforth, we call the former ’global group integrity’ and the latter ’local group integrity’). As mentioned above, global group integrity implies the interaction between the global reference and each individual. This interaction can be understood as a mean-field approximation in Ising spin glass models. Mean field approximation describes a many-body problem by analysing a model where the local interaction between neighbouring microscopic spins is replaced by an interaction with an average spin called the mean-field [79,92,93]. Obviously, our definition of the global reference also shares the framework of the mean-field approximation. In general, when there are a sufficiently large number of individuals in a system, interactions between individuals can be approximated as the asymptotic average corresponding to the bottom-up interactions in the limit (self-consistent method in the mean-field). This approximation is assured by the assumption that interactions are homogeneous in a sufficiently large system.

However, the mean-field assumption would not necessarily be relevant for a small system where interactions are not homogeneous. The heterogeneity of interactions is not negligible in small fish schools. In this sense, the difference between the global and the local group integrities would be meaningful in terms of the measure of the heterogeneous interaction. If the global group integrity tends to show more significance than that of the local group, the local interaction would be over-estimated in terms of the causal power. In contrast, if the local group integrity tends to show more significance than the global group, the local interaction would be under-estimated. If both values are approximately equal, the approximation can work in terms of the causal power.

For the above-mentioned reasons, we compare the peak 〈Φ(N)〉 values for all the parameter settings in each timescale. Figure 8 shows the comparison between the peak 〈Φ(N)〉 values (top 20 for each) selected from the heat maps along with the timescales (the number of top *x* never affects our result. See Appendix A). Each of these 〈Φ(N)〉 values were selected from Figure 4 (Appendix A) and Figure 8 (Appendix A). For clarity, we denote $\Phi_{local}$ (the local group integrity) for the local parameters and $\Phi_{global}$ (the global group integrity) for the global parameters. $\Phi_{local}$ is an intrinsic causal power generated by the bottom-up interactions, whereas the $\Phi_{global}$ is based on the interaction with the global reference. The main concern is the ability of the system to generate a causal power within different parameter settings. For a longer timescale, we previously confirmed that Φ(N) values for both parameter settings generally increase; however, beyond a specific timescale, they eventually saturate. Figure 8 shows how each 〈Φ(N)〉 value changes with timescales. The vertical axis represents the difference between $\Phi_{local}$ and $\Phi_{global}$, whereas the horizontal axis represents timescales. The difference between the two values gradually shifts from negative to positive, and finally approaches zero. In other words, in the long time scale, the local interaction can be approximately homogeneous in terms of the group integrity.

We also confirm the outstanding property of a school of five fish at Δt=20/120 s. Interestingly, this timescale approximately corresponds to the reaction time of the fish (e.g., *Hemigrammus rhodostomus* [42] and *Plecoglossus altivelis* in Appendix A). The graph shows that the difference between local and global group integrity is the most significant when the group size is five-fish at Δt=20/120 s compared with other group sizes (Table 1). Owing to the local group integrity that is based on the bottom-up interaction, the peak shows that the degree of the global interactions underestimates the local interaction with respect to the 〈Φ(N)〉 values. In other words, the group integrity generated by the local interaction has considerable causal power than assuming their homogeneous interaction. In this sense, from the IIT 3.0 perspective, the heterogeneous interaction becomes the most remarkable in the five-fish school. No remarkable difference was observed among two-, three-, and four-fish schools (Table 1). Only the school of five fish showed strong local group integrity.

### 2.4. Correlation between 〈Φ(N)〉 and $\sigma^{2}\left(\Phi \left(N\right)\right)$

It is worth noting the positive correlation between the mean Φ values and its variance in a real school, especially in the local interaction systems. Figure 9 plots all 〈Φ(N)〉 values and their Fano factors $\sigma^{2}\left(\Phi \left(N\right)\right)$/〈Φ(N)〉 (normalised variance) of all parameter settings. Those of the global parameter settings show the same result (Appendix A), but show a relatively weak tendency compared with the local systems. The strong correlation between 〈Φ(N)〉 and its variance in real fish bears critical implications. This correlation indicates that the schools with high 〈Φ(N)〉 values contain two extremes: low Φ(N) values and high Φ(N) values (recall each collective state has a unique Φ(N) value and 〈Φ(N)〉 represents the average of those $2^{n}$Φ(N) values for *n*-fish school). The real school systems tend to have low Φ(N) values even if the group integrity (i.e., average Φ(N) values of all collective states) is significant. We also point out that the pure stochastic model, such as that generated using Markov process, never shows such a tendency (its variance is approximately 0.4 at the most [77]). Their variances remain low even if the 〈Φ(N)〉 increases. The high variances in the highly integrated system are one of the characteristic properties observed in a real school.

### 2.5. New Classification of Schools as Different Autonomous Systems

In this section, we relate our IIT 3.0 analysis to the real group behaviours to show that each group size can be classified as a different autonomous system. The previous analysis (in Section 2.2.2, Section 2.2.3 and Section 2.3) reveals that there are specific parameter settings which pick up the (statistically significant) characteristics in terms of the high average group integrity 〈Φ(N)〉. For instance, the low visual field parameters regions are for two-fish; the high visual field parameters regions are for four- and five-fish. This section aims to unveil what are the corresponding group behaviours (or information process) in these regions.

To provide a correspondence to the actual group behaviour, we tried to analyse the detail of Φ values of the single-OFF collective state (only one individual is in the OFF state and the other members are in the ON state). The main reason for this selection is, in high 〈Φ(N)〉 regions, more than 99% of collective states are made up of only two collective states, that is, all ON state and the single-OFF state. All ON state collective states simply indicate the group cohesion. Meanwhile, the single-OFF collective states represent only one fish in the group behaves differently (non-changing direction or the head of the group). Through the analysis of the single-OFF collective states, we can estimate how individual actions (a single-OFF state) affect the entire system.

Here we define two kinds of matching rate to examine the correspondence between a single-OFF state fish and group formation or group information. By considering both sides of roles of the single-OFF state fish, we can relate the information flow to the group formation. First, we define ‘MIP-cut matching rate’ to examine the inter-subgroups information process. To get this value, we can simply calculate the proportion of the event that MIP cut lies between a single-OFF state and the other ON state (when the collective state 1011 is given, the MIP cut is {2}⇏{1,3,4} or {1,3,4}⇏{2}). The MIP-cut rate indicates how the OFF state fish are involved in the entire information process of the system. The high MIP-cut matching rate is a necessary condition for the leadership condition (ii), but not a sufficient one. The one-way of information flow (i.e., feed-forward structure) finally determines the leadership condition (ii).

Second, we define ‘positional leader (PL) matching rate’, which shows that a single OFF-state fish highly correlates with positional leadership (leadership property (i)). In this study, the positional leader is defined as the individual in the first position with respect to the average group direction. We can compute the rate at which single OFF-state individuals actually correspond to the positional leader. Let N and T be sets {1,2,…,N} and $\left\{1,2,\ldots ,T_{max}\right\}$, respectively, where $T_{max}$ is the maximum time for a given time step. Then, the single OFF state function is ${\mathrm{OFF}}_{\mathrm{single}}\left(t\right):T\to N⋃\left\{0\right\}$. This function returns the index of the OFF-state individual when the collective state has only one OFF state; otherwise, it returns 0. The positional leadership function is $P_{\mathrm{Leader}}\left(t\right):T\to N$. This function selects the index of the positional leader.
(12)$$\mathrm{PL}\;\mathrm{Match}\;\mathrm{Rate}\;\left(\%\right)=\frac{\left|\{t|{\mathrm{OFF}}_{\mathrm{single}}\left(t\right)=P_{\mathrm{Leader}}\left(t\right),t\in T\}\right|}{\left|\{t|{\mathrm{OFF}}_{\mathrm{single}}\left(t\right)\in N,t\in T\}\right|}\times 100$$

The above equation (Equation (Equation 12)) indicates the frequency of occurrence of the OFF state individual as the positional leader.

Table 2 shows the result of the MIP cut matching rate and PL matching rate for selected parameter settings. The basic trend of PL matching rate becomes higher when $\xi_{VF}<2\pi$. In other words, positional leadership corresponds to the single-OFF collective state. However, for $\xi_{TR}>0$, the single-OFF collective state contains another situation such as the weak direction changing of the OFF state fish (Appendix A). These events decrease the PL matching rate for four- and five-fish schools, but overall their PL matching rate keeps a high rate (over 70 %: when $\xi_{TR}=0$, much more correlation can be observed in Appendix A). Meanwhile, the poor MIP cut matching rate is observed only in three-fish schools for all parameter settings (Table 2 and Appendix A). These low values in three-fish schools indicate that the fish behaviour unsuccessfully connects with the group information process.

Before we get into the details of the analysis of the classification, we review the meaning of the relationship between the Φ value (group integrity) and the MIP cut. As noted in Section 2.1, the MIP cut was for the weakest link in the system, and the Φ value represented the degree of information loss due to the MIP cut. As Oizumi et al. have already proposed [57], when the value of Φ is low, the system has a feed-forward structure. Meanwhile, when the Φ value is high, the opposite flow is also high, indicating that the system has a feed-back structure. We also point out that, in actual living systems, it is hard to explicitly distinguish between the feed-back and feed-forward structure. In the real system, their relationship is rather a relative one. While considering these relative relationships, we classify the system into different autonomous systems according to the group sizes. Notably, the cohesion of the group is ensured for all parameter settings used in this analysis (Table 3). The event which one individual is far from the group rarely happens.

(I) The two-fish school (Figure 10a) as “chasing”: each Φ value increases with the number of ON states in the collective states. One of the characteristic point of the two-fish school is that the MIP cut always lies between two fish by its definition. As we have confirmed in Table 2, this cut also corresponds to the positional leadership. Φ values in Figure 10a indicate that the feed-back structure can be confirmed in all ON states and the feed-forward structure can be confirmed in a single OFF state and all OFF states. This feed-forward structure in a single OFF state implies that it satisfies the leadership characteristic that we defined as the existence of asymmetric information flow (property (ii)).

However, some unignorable differences between the two- and four-fish’s leadership exists. The main difference is the narrow visual field in the two-fish school. Unlike a wide visual field, but similar to the four-fish school, the visual field of two-fish’ is restricted approximately from $\xi_{VF}=0.18\pi$ to $\xi_{VF}=0.54\pi$ (rad). Furthermore, the distribution of Φ values in the visual field is not symmetrical (Figure 4b and Figure 5b). The existence of all OFF states is also another characteristic of the two-fish school because in other group sizes, all OFF states never occur at such a high rate. Therefore, we should distinguish the system of a two-fish school from that of a four-fish school. We tentatively refer to this system as ‘chasing’ because the low $\xi_{VF}$ in the two-fish school shows one fish following the other within their narrow visual field.

(II) The three-fish school (Figure 10b) as ‘fission-fusion’: The characteristics of the three-fish school are its weak correspondence of the MIP cut and the positional leader (even if the blind spot exist in the three-fish school, the system never shows the correlation between the MIP cut and positional leader (Table 2) and Appendix A). The MIP cut never determines the position of the single OFF state individually, directionally, and informationally (the right figure inset of Figure 10b). All we can confirm here is that the group integrity increases when the group makes cohesion (all ONE state), whereas it decrease when one fish is not in the group (a single-OFF state). We consider that the term ‘fission—fusion’ sums up this situation well. Although the term of ‘fission-fusion’ already exists in collective behaviour, its essence is in the collective process of merge and split dynamics [94,95,96,97]. We consider that the dynamics we are analysing here are not very far from their interest.

(III) The four-fish school (Figure 10c) as ‘leadership’: from Table 2, the correlation between the MIP and the positional leader increases from this group size (especially when $\xi_{TR}=0$). Furthermore, Figure 10c indicates that Φ values of the single-OFF collective state show lower than those of the all ON states. The single-OFF collective state shows the feed-forward structure (the right figure in Figure 10c). Thus, the four-fish school satisfies two leadership conditions as we have confirmed in [77].

We point out that the ‘leadership’ we used here is different from the traditional leaderships. First, the relation between the leader and its followers is not a one-to-one correspondence (the leader and its follower) similar to transfer entropy [42,43,44,49], but a one-to-many relationship induced by the MIP cut (the leader and the rest of its members: the relation similar to {1}⇒ {2, 3, 4}). The leadership in IIT 3.0 never determined as the pair of individual relation, but as the system divided into two sub-groups.

Five-fish school (Figure 10d) as ‘Interactive’: Five-fish schools also show the same trend as that of four-fish schools. Both MIP-cut points correlate with the positional leadership. The Φ values of all ON collective states are significantly larger than those of single-OFF collective states. As a result, both causal structures become isomorphic. Nevertheless, we insist that these two group sizes are quite different systems for two reasons that are discussed below.

First, the functional role played by an OFF state individual in the system is different for four- and five-fish schools. When the group size is four, the Φ values at $\xi_{TR}$=0 are more significant than those at other parameters (Appendix A), which suggests that the OFF state individuals in the system represent an interaction that depends mainly on the field of view and not on the tuning rate. On the other hand, when the group size is five, the system shows the same trend of Φ values even if $\xi_{TR}$ is more than 0 (Appendix A). In these parameter regions, an OFF state has two meanings, namely “out of visual field” and “non-turning”, in the group. In other words, the action of the individual (non-turning) has high causal values after exceeding five-fish schools. Second, the local integrity exceeds the global integrity only in case of the five-fish schools. This trend indicates the qualitative difference between four- and five-fish schools. As we have noted in Section 2.3, the difference represents the heterogeneity of the interactions among the individuals. Therefore, this trend may have some connections with above-mentioned first reason because the action of the individual starts to have a definite causal meaning.

For these reasons, we believe that four- and five-fish schools should be distinguished. Therefore, we would like to refer to this system as ‘interactive’ to distinguish from other systems because, at this phase, the individual’s behaviour starts showing the strong causal power in the group.

## 3. Discussion

The IIT analysis for the collective behaviour forces us to shift our focus from an external perspective (information transfer) to an internal perspective (internal causal structure). The former represents the events occurring in the system while the latter represents the functions of the system [51,57,69,70]. Hitherto, the latter approach, from an internal perspective, barely applies to collective behaviour of animals. Transfer entropy is a representative method of the former approach, especially in a small group [42,43,44,49]. The transfer entropy approach first constructs by selecting a pair of individuals from the given group, it then measures the degree of information and misinformation [42], and finally determines the leadership relation in the group. From the external perspective, the leadership in the group is itself a ubiquitous property. All these approaches are aimed to reveal the properties of the system behaviour of the group.

The importance of our approach is mainly based on two aspects: the shift from an external to an internal perspective, and a tentative answer to the question, ‘what is the criterion of the group?’. Next, these aspects are discussed in detail.

First, the representative analysis from the external perspective is the transfer entropy analysis [42,43,44,49]. As discussed earlier, this method is constructed by selecting a pair of individuals from the group; however, this method only represents the predictability of each pair of individuals from their external behaviours in a series of events.

The analysis based on the internal perspective provides us with different perspectives. The cellular automaton study in [51] could help in understanding this phenomenon. In this study, although IIT 3.0 can distinguish from Classes I to IV, the cell number for their analysis was restricted to six. This number is significantly low to distinguish the external cell behaviour with respect to their corresponding rules; therefore, there is no room here to determine the system using the very little information of the cells’ behaviour. The distinction becomes possible only if the cellular automaton’s intrinsic causal structures of IIT 3.0 are considered, but not their external behaviour.

The distinction between external and internal perspectives might be vague. This is because of the correspondence between the IIT-induced leadership (i.e., feed-forward information by the MIP) and positional leadership in the group. Although IIT 3.0’s analysis never refers to the external behaviour, but the transition of collective states, the correlation between the IIT 3.0 analysis and its external behaviour makes the external and internal perspective vague. In contrast, as we have already confirmed, the MIP-cut does not always correspond to the positional leader such as the three-fish school, even if the blind spots are introduced. In other words, the correlation between the internal and external perspective is not always self-evident. Furthermore, our previous study indicates that the transfer entropy approach does not distinguish the leadership as an autonomous system [77].

Throughout our analysis, we have confirmed that each group size has a unique characteristic causal structure via IIT 3.0 analysis by extending our previous method. Considering the long time scale up to Δt=120/120 enables us to discover some outstanding properties that were not observed in our previous study. For instance, the ‘chasing’ for two-fish schools is only confirmed in the long timescale (see Figure 3 and Appendix A). We indicate that this ‘chasing’ property is not attributable to the calculation method used in IIT 3.0. In our other study, we investigated the group integrity as a sub-group (e.g., two-fish sub-schools, three-fish sub-schools, and four-fish schools etc.); the two-fish subgroup in the larger school never showed a distribution similar to genuine two-fish schools [98], that is, the ‘chasing’ is never confirmed in other subgroups (other group sizes as well). The group integrity for each group size sufficiently reflects their internal causal structure. The same trend applies the “fission–fusion” in the three-fish school; however, the difference between four- and five-fish schools is relative (only the difference in the position of peak Φ values).

Among the four characteristics (chasing, fission–fusion, leadership, and interactive), ‘interactive’ (five-fish school) has much to be considered because the effect of the calculation method of IIT 3.0 for a specific group size cannot be rejected; however, our timescale analysis helps to distinguish four- and five-fish schools as autonomous systems. As discussed in Section 2.3, comparing the global group integrity with the mean-field method, we can evaluate the interaction heterogeneity in terms of the group integration. First, it turned out that the degree of the interaction homogeneity can be observed for all group sizes in the long timescales. This result ensures that the same kind of approximation done in mean-field also works in the group integrity framework: the bottom-up and the top-down processes in the group match in the long timescale. Second, around the reaction time of *Plecoglossus altivelis*, such an approximation does not work. Especially, compared with other group sizes, the five-fish school exhibits the outstanding property that the local group integrity exceeds the global group integrity. This trend indicates that the average approximation underestimated the causal power of the local interaction. The local interaction generates more group integrity than in case of homogeneous interactions. In the result of Section 2.5, we also point out that this heterogeneity shows some resonances (the action of an individual starts to have strong causal power). Although this view still might be speculative, it gives some hints for future study.

So far, we have attempted to analyse the actual data of *Plecoglossus altivelis*’s collective behaviour. Here, the results of our classification, which will produce similar results in other species, are unknown. Accordingly, we believe that our classification has its own values in the context of the analysis of other diverse species in the future studies. However, what the causal analysis we have attempted here seems to have one suggestion of itself. In the second half of this section, we will discuss the theoretical implications of applying IIT 3.0 to real collective behaviour.

Our IIT analysis led us to reconsider the fundamental question: ‘what is the criterion for forming a group?’. This question is rarely asked, but simply assumed. Several examples can be considered. The information transfer in the group, for instance, assumes the individual’s collective behaviour as one collection [38,42]. The other example can be seen in the criticality arguments [13,14,15]. The analysis of the scale-free correlation in the birds can only be possible when we define the average values (for example mean directions or speeds). Of course, these evaluation methods can be applied even if the groups are segmented; however, the computation of these average values implicitly assumes that an observer considers the set of birds as one collective (for example, the flock as an analysis object can be likened to one brain and one mind [13,27]).

We do not insist that their assumption on the group’s criterion is itself inappropriate; however, this assumption seems inevitable for group analysis. A sufficient number of individuals is likely to satisfy those criteria. However, in the case of a small group similar to the transition from two-to five-fish schools, this problem is not negligible because there is still no consensus on the group criterion for such sizes. The question of what kind of the difference exists among these small groups has remained unanswered to date.

We consider that the heap paradox would help to shape the core of this problem [30]. The heap paradox is constructed by sequentially adding each grain on the floor when the aggregation of grains becomes a heap. The key of this paradox is in the fact that the same successive procedure (i.e., grain dropping) can make the system different, qualitatively. The paradox asks us when does qualitative difference begin and how can we know that. This paradox can be observed all over the domain of complex systems concerned with parts and entire relations (e.g., Neurons-build consciousness [56,57,60,61] and individuals’ group building behaviour [27,34,99]). The paradox overlaps our arguments of collective behaviour on the topic: ‘the criterion of the group behaviour’.

The clue for understanding the heap paradox is in the way the grain is distributed. To understand this, it is sufficient to confirm that the random distribution of the grain never constructs the heap; rather, the heap is constructed by the distribution of flat grain. In other words, the agent doing the same procedure inevitably commits his/her decision on where to drop the grain, anticipating the forthcoming unbuilt heap. The heap paradox implicitly includes the agent’s commitment to grain droppings. The paradox begins when we forget the agent’s ontological commitments.

The agent’s commitment can be considered in the context of a manipulative view in IIT 3.0. As Tononi claims, ‘something can be said to exist only if it has cause–effect power’ [61]; the causal–effect structure of an autonomous system represents ‘what causes what’, considering all counterfactual states. Lombardi et al. correctly indicated that the central idea of ‘differences that make differences’ represents the potential intervention (manipulation) in the cause–effect structure of the system [71]. We can also apply this argument to collective behaviour. The intrinsic causal structure in a school represents how the fish virtually affects one another to maintain an aggregation. The manipulative view in IIT 3.0 involves the agent’s commitment to the intrinsic causal perspective.

Now, we are ready to present the two tentative answers to the paradox from IIT 3.0: from the inner-system perspective and the inter-system perspective.

Figure 9 corresponds to the former answer. As we had discussed earlier, the strong correlation between 〈Φ(N)〉 and Fano factor $\sigma^{2}\left(\Phi \left(N\right)\right)/\langle \Phi \left(N\right)\rangle$ is innate for a real school (no observation in the artificial condition, such as the random Markov networks.) This correlation indicates that the high average group integrity in a real school has both low and high Φ values, simultaneously. In other words, the real school groups coexist as tightly mutual connections with feed-back (united: high Φ) and weak connections with no feed-back (separated: low Φ) individuals that exist in the same fish school. Unlike other methods, the presence of multiple Φ(N) values in the same group is critical in IIT 3.0. This coexistence in IIT 3.0 can prevent the paradox’s assumption because a law that excluded the middle (i.e., any proposition must be true or true for its negation: in this case, we mean that the grains would be united or not) becomes invalid. This is one possible answer to the paradox.

The latter answer (inter-system) can be obtained from the classification of schools as autonomous systems. Our finding is that the difference in group size differs with the autonomous system. This classification itself makes the paradox’s assumption invalid because the properties of the group change according to the group’s size from IIT 3.0 perspective. There is no definitive point where the group behaviour begins, but its definition may be different for the way of forming the group in a given environment. In other words, the question of when the school begins becomes invalid. This is another possible answer to the paradox.

We investigated the schools. By applying IIT 3.0, there are possibilities for new approaches to the problem of the collective behaviour, which remains unsolved. We believe that our approach elucidates on not only the collective behaviour but also the application of IIT 3.0.

## 4. Materials and Methods

### 4.1. Ethics Statement

This study was carried out in strict accordance with the recommendations in the Guide for the Care and Use of Laboratory Animals of the National Institutes of Health. The protocol was approved by the Committee on the Ethics of Animal Experiments of the University of Tsukuba (Permit Number: 14-386). All efforts were made to minimise suffering.

### 4.2. Φ Computation

All computations, in this study, were performed using the PyPhi software package with the CUT_ONE_APPROXIMATION to Φ.

### 4.3. Experimental Settings

We studied *ayus* (*Plecoglossus altivelis*), also known as sweetfish, which are common and widely bred in Japan. Juvenile *ayus* (approximately 7–14 cm in body length) display typical schooling behaviour, though adult *ayus* tend to show territorial behaviour in environments where fish density is low. We purchased fingerlings from Tarumiyoushoku (Kasumigaura, Ibaraki, Japan) and housed them in a controlled laboratory. Approximately 150 fish lived in a 0.8 m${}^{3}$ tank of continuously filtered and recycled fresh water with temperature maintained at 16.4 °C. They were fed with commercial food pellets. Just before each experiment was conducted, fish were randomly chosen and separated to form a school of each size and were moved to an experimental arena without pre-training. The experimental arena comprised a 3 × 3 m${}^{2}$ shallow white tank. The water depth was approximately 15 cm so that schools would be approximately 2D. The fish were recorded with an overhead grey-scale video camera (Library GE 60; Library Co. Ltd., Tokyo, Japan) at a spatial resolution of 640 × 480 pixels and a temporal resolution of 120 fps.

### 4.4. Data Summary

The definition of the cohesion rate in Table 3 is as follows: We compute the distance for all pairs of fish at time *t*. The group in cohesion implies all pairs of distances within a certain length (in this case, we set 400 (mm). The figures in the parentheses are 700 (mm)). The cohesion rate is the rate of cohesion throughout all time steps. The data indicates that the group of all data sets are in a cohesion state throughout our experiment.

The analysis in this study is done in enough time length, but with relatively small sample sizes. As Collignon et al. had suggested [100], the inter-group variability was also observed for *ayu*; as our analysis had shown, the variances of Φs within the same group size were not so large (in Figure 10). Besides, our previous studies showed that the switching time of positional leadership obeys the power-law distribution for all datasets [101]. In this sense, the statistical properties of our dataset can regard the approximately same in terms of one system. Hence, we consider the inter-group variability in this study never affects our results critically.

### 4.5. Timescales

We prepared the trajectory data at different time intervals (In Python X[::δt]) and applied smoothing by convolving three data points to reduce noise. δt is a timescale in δt/120.

### 4.6. The Definition of ON and OFF State for Each Parameter

We define a function for each parameter that returns either 0 (OFF) or 1 (ON), for any given input value. Generally, we denote a function as $F_{i}^{t}\left(\cdot \right)$, where *F* is the name of the function, *i* is the index of the individual, and *t* is the time. The arguments of the function can be either in the position vectors, $x_{i}\left(t\right)$, or the velocity vectors, $v_{i}\left(t\right)$, of each individual, at time *t*. In general, the dimensions of these vectors are d≤3; the experimental setup used here gives d=2, and the number of individuals is *N*.

#### 4.6.1. Local Parameters

Distance function $D_{i}^{t}\left(x_{1}\left(t\right),x_{2}\left(t\right),\cdots ,x_{n}\left(t\right)\right)$: $R^{d}\times R^{d}\times \cdots \times R^{d}\overset{}{\to }\left\{0,1\right\}$For each individual, *i*, we obtain a set $S_{i}^{t}=\left\{j|d\left(x_{i}\left(t\right),x_{j}\left(t\right)\right)<\xi_{D},j\neq i\right\}$ of all other individuals within a specified distance, $\xi_{D}$. Here d(x,y) gives the Euclidean distance between x and y. Then, $D_{i}^{t}\left(x_{1}\left(t\right),x_{2}\left(t\right),\ldots ,x_{n}\left(t\right)\right)=1$ when $|S_{i}^{t}|>0$ and is 0 otherwise, where |S| denotes the number of elements of the set, *S*.Visual field function $B_{i}^{t}\left(x_{1}\left(t\right),x_{2}\left(t\right),\ldots ,x_{n}\left(t\right),v_{1}\left(t\right),v_{2}\left(t\right),\cdots ,v_{n}\left(t\right)\right):R^{d}\times R^{d}\times \cdots \times R^{d}\overset{}{\to }\left\{0,1\right\}$For each individual we form the set $O_{i}^{t}=\left\{j\right|$ arg($v_{i}\left(t\right)$, $x_{i}\left(t\right)-x_{j}\left(t\right)$) $<\xi_{VF}$, j≠i} of all other individuals whose velocity vectors point in a direction within an angle $\xi_{VF}$ of the focal individual. The function arg($x_{1}\left(t\right)$, $x_{2}\left(t\right)$) gives the angle between the two vectors. Then, $B_{i}^{t}\left(x_{1}\left(t\right),x_{2}\left(t\right),\ldots ,x_{n}\left(t\right),v_{1}\left(t\right),v_{2}\left(t\right),\cdots ,v_{n}\left(t\right)\right)=1$ when $|O_{i}^{t}|>0$ and is 0 otherwise.Turning rate function $T_{i}^{t}\left(v_{i}\left(t\right),v_{i}\left(t-\Delta t\right)\right):R^{d}\times R^{d}\overset{}{\to }\left\{0,1\right\}$The turning rate function returns 1 when an individual’s turning rate exceeds a specified threshold, δ. That is, $T_{i}^{t}\left(v_{i}\left(t\right),v_{i}\left(t-\Delta t\right)\right)=1$ when arg($v_{i}\left(t\right)$, $v_{i}\left(t-\Delta t\right))\geq \xi_{TR}$ and is 0 otherwise. The time step used in this study is from Δt=0.05 to Δt=1.0 s.To obtain the states of the school, we take a conjunction of this result, that is, $D_{i}^{t}\left(x_{1}\left(t\right),x_{2}\left(t\right),\cdots ,x_{n}\left(t\right)\right)\wedge B_{i}^{t}\left(v_{1}\left(t\right),v_{2}\left(t\right),\cdots ,v_{n}\left(t\right)\right)\wedge T_{i}^{t}\left(v_{i}\left(t\right),v_{i}\left(t-\Delta t\right)\right)$ for each individual, *i*. The conjunction is given as $\wedge :{\left\{0,1\right\}}^{2}\overset{}{\to }\left\{0,1\right\}$ where 1∧1=1 and is 0 otherwise. Thus the state of each individual *i* at time *t* is $s_{i}\left(t;\xi_{D},\xi_{VF},\xi_{TR}\right)\in \left\{0,1\right\}$ which depends on the three parameter values $\left(\xi_{D},\xi_{VF},\xi_{TR}\right)$. The state of the school at time *t* is a vector $s\left(t\right)=\left(s_{1}\left(t\right),s_{2}\left(t\right),\ldots ,s_{n}\left(t\right)\right)\in {\left\{0,1\right\}}^{n}$, where the parameter dependence has been omitted for simplicity.

#### 4.6.2. Global Parameters

Average direction function $Avd_{i}^{t}\left(V\left(t\right),v_{i}\left(t\right)\right):R^{d}\times R^{d}\overset{}{\to }\left\{0,1\right\}$V(t) is the average of $\left\{v_{1}\left(t\right),v_{2}\left(t\right),\ldots ,v_{n}\left(t\right)\right\}$. If an individual’s direction of motion deviates from the average by more than a threshold amount, $\Xi_{AD}$, then the individual is in the OFF state: that is, $Avd_{i}^{t}\left(V\left(t\right),v_{i}\left(t\right)\right)=1$ when arg(V(t), $v_{i}\left(t\right))\leq \Xi_{AD}$, and is 0 otherwise.Centre of mass function $Com_{i}^{t}\left(X\left(t\right),x_{i}\left(t\right)\right):R^{d}\times R^{d}\overset{}{\to }\left\{0,1\right\}$X(t) is the average of $\left\{x_{1}\left(t\right),x_{2}\left(t\right),\cdots ,x_{n}\left(t\right)\right\}$. If an individual is further from X(t) compared with a specified threshold $\Xi_{CM}$ then the individual is in the OFF state: that is, $Com_{i}^{t}\left(X\left(t\right),x_{i}\left(t\right)\right)=1$ when d(X(t), $x_{i}\left(t\right))\leq \Xi_{CM}$ and is 0 otherwise.To obtain the state of the school, we take a conjunction of these results to obtain a state for each individual, which depends on the pair $\left(\Xi_{AD},\Xi_{CM}\right)$:, $s_{i}\left(t;\Xi_{AD},\Xi_{CM}\right)=Avd_{i}^{t}\left(V\left(t\right),v_{i}\left(t\right)\right)\wedge Com_{i}^{t}\left(X\left(t\right),x_{i}\left(t\right)\right)\in \left\{0,1\right\}$. The state of the school at time *t* is then a vector $s\left(t\right)=\left(s_{1}\left(t\right),s_{2}\left(t\right),\ldots ,s_{n}\left(t\right)\right)\in {\left\{0,1\right\}}^{n}$, where the parameter dependence has been omitted for simplicity.

## Figures and Tables

**Figure 1 entropy-22-00726-f001:**
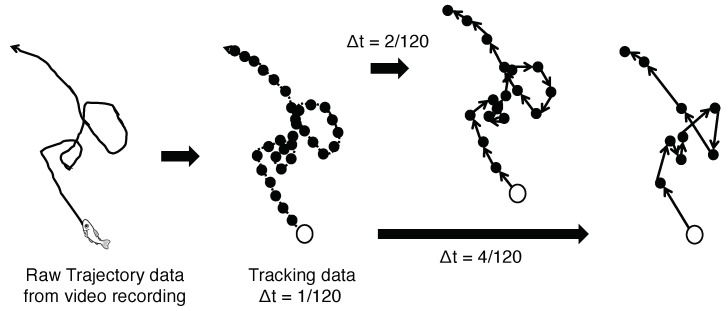
Timescale rules. The video recordings present each trajectory. The frame rate of our device is 120 fps. The velocity vectors of the time scale, Δt, define every Δt step-length when the time scale is Δt=δt/120. The timescales are Δt=5/120, Δt=10/120, Δt=20/120, Δt=40/120, Δt=80/120, and Δt=120/120 s. Long-time scales eliminate subtle noise-like movements in a school (or support the predictability of other fish’s movements).

**Figure 2 entropy-22-00726-f002:**
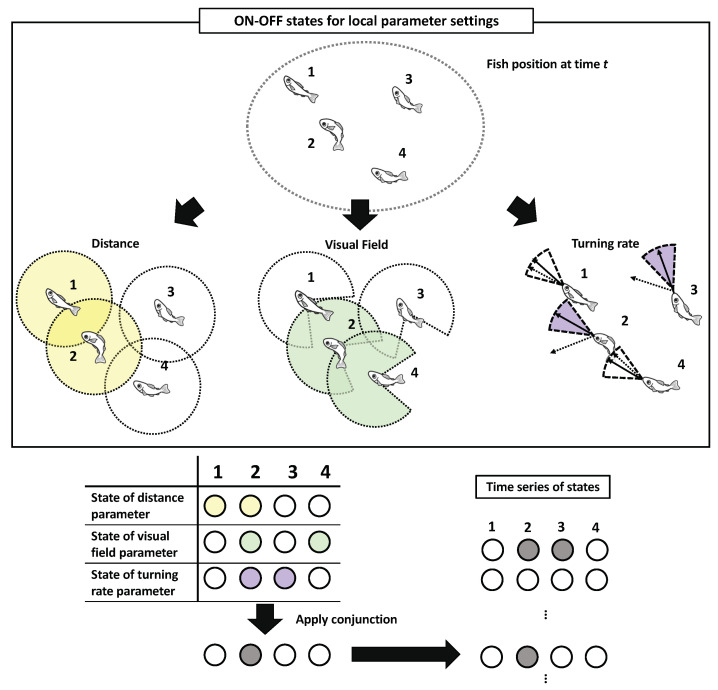
The definition of ON and OFF states for local parameter settings. Three parameters are used to determine a school’s state (Yellow: distance, Green: visual field, and Purple: turning rate). The coloured individuals are in the ON state. We take a conjunction of the three school states to obtain the final school state at time *t*. Subsequently, we compute Φ from a time series of these states using PyPhi. The meanings of the above figures is as follows: Distance means fish 1 and 2 are in the ON state because their interaction radius includes each other. Visual Field means fish 2 (fish 4) is in the ON state because fish 1 (fish 2) is included in its visual field. Turning rate means the sector represents the threshold of this parameter. The bold vector represents the velocity vector at time t−1 and the dotted vector represents the velocity vector at time *t*. Fish 2 and 3 are in the ON state because of the dotted direction out of their own threshold sector.

**Figure 3 entropy-22-00726-f003:**
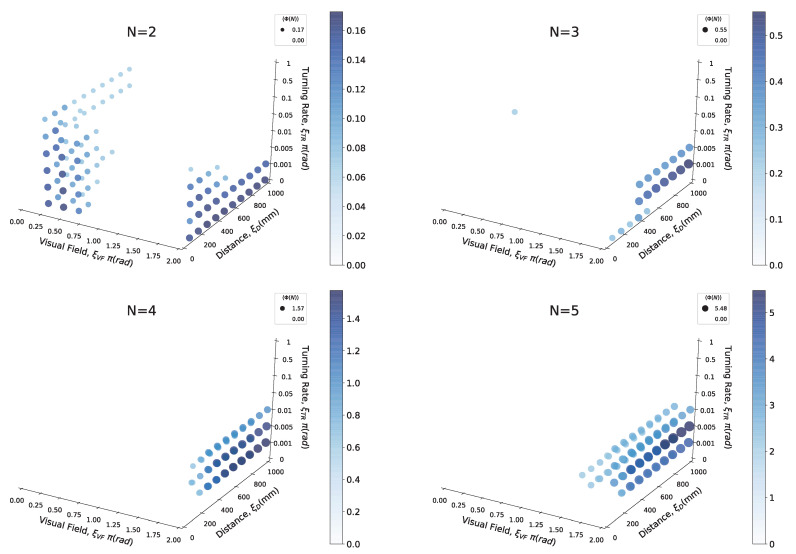
Three-dimensional distribution of the mean 〈Φ(N)〉 values with respect to the three parameters according to the group size for all experimental data (Δt=20/120 s). The ball size and shaded colours represent the 〈Φ(N)〉 strength. Owing to visibility, we only show the points over 0.4*${\langle \Phi \left(N\right)\rangle }_{MAX}$, where ${\langle \Phi \left(N\right)\rangle }_{MAX}$ implies the 〈Φ(N)〉 values of the maximum cell for each group size. We measured the 〈Φ(N)〉 values over the main complexes and full subsystems throughout our analysis. All the graphs only refer to the main complexes. For similar distributions, please refer to [77]. Appendix A include all the mean 〈Φ(N)〉 and mean $\sigma^{2}\left(\Phi \left(N\right)\right)$ values for all the timescales.

**Figure 4 entropy-22-00726-f004:**
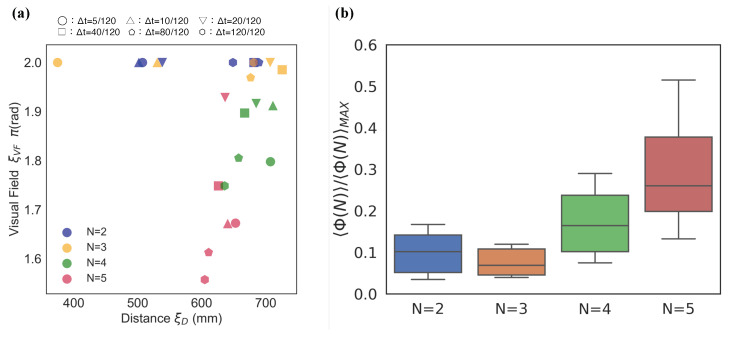
(**a**) The mean distribution over $\frac{1}{2}{\langle \Phi \left(N\right)\rangle }_{MAX}$, where $\xi_{VF}>\pi$ and $\xi_{TR}<0.05$. Each colour corresponds to the group size, and each shape corresponds to the time scale: Δt=5/120 s (circle), 10/120 s (upward triangle), 20/120 s (downward triangle), 40/120 s (rectangle), 80/120 s (pentagon), and 120/120 s (hexagon). ${\langle \Phi \left(N\right)\rangle }_{MAX}$ represents the maximum 〈Φ(N)〉 value of all the cells in the 〈Φ(N)〉 distribution. The distributions of N=2 and N=3 for all the scales distributed on the complete visual fields ($\xi_{VF}=2\pi$). In contrast, the distributions of N=4 and N=5 for all the scales distributed on the lower right field ($\xi_{VF}<2\pi$). (**b**) The box plot for the mean normalised 〈Φ(N)〉 values, where $\pi <\xi_{VF}<2\pi$ for all datasets. Each datum is divided by the ${\langle \Phi \left(N\right)\rangle }_{MAX}$ in the region of $\pi <\xi_{VF}<2\pi$ and $\xi_{VF}<\pi$ (this graph uses all the 〈Φ(N)〉 values: no restriction such as $\frac{1}{2}{\langle \Phi \left(N\right)\rangle }_{MAX}$). The 〈Φ(N)〉 values of four- and five-fish schools are significantly higher than those of two- and three-fish schools. For comparison, Appendix A presents the same box plot for high turning rates of $\xi_{TR}\geq 0.05$.

**Figure 5 entropy-22-00726-f005:**
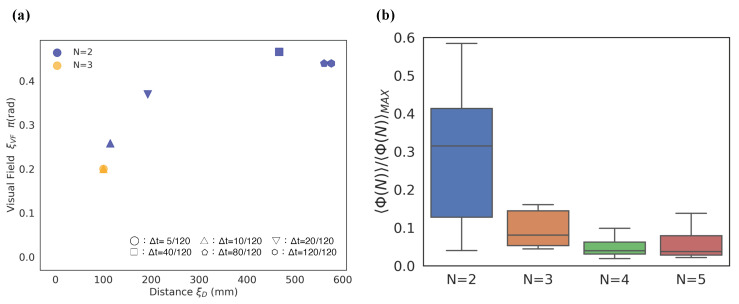
(**a**) The mean distribution over $\frac{1}{2}{\langle \Phi \left(N\right)\rangle }_{MAX}$, where $\xi_{VF}<\pi$ and $\xi_{TR}<0.05$. Each colour corresponds to the group size, and each shape corresponds to the time scale: Δt=5/120 s (circle), 10/120 s (upward triangle), 20/120 s (downward triangle), 40/120 s (rectangle), 80/120 s (pentagon), and 120/120 s (hexagon). No value exceeds $\frac{1}{2}{\langle \Phi \left(N\right)\rangle }_{MAX}$ for the four- and five-fish schools. Some of the three-fish school exceed this value; however, only in a few samples. (**b**) The box plot for the mean normalised Φ values, where $0<\xi_{TR}<\pi$ for all datasets. The data were divided with the maximum ${\langle \Phi \left(N\right)\rangle }_{MAX}$ in $0<\xi_{VF}<\pi$ and $\xi_{TR}<0.05$ (this graph uses all the 〈Φ(N)〉 values: no restriction such as $\frac{1}{2}{\langle \Phi \left(N\right)\rangle }_{MAX}$). The 〈Φ(N)〉 values in the two-fish school were determined as ‘chasing’, which is the opposite of ‘leadership’. For comparison, Appendix A depicts the box plot for the high turning rate, $\xi_{TR}\geq 0.05$, under the same condition. The statistical test is included in Appendix A.

**Figure 6 entropy-22-00726-f006:**
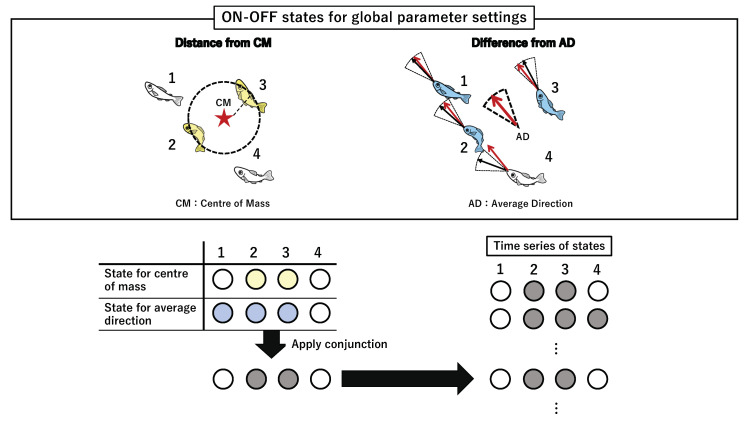
Definition of ON and OFF states for global parameter settings. Two parameters determine a school’s state (Yellow: Centre of Mass and Blue: Average Direction). Coloured individuals are in the ON state. We calculate the conjunction of the two school states and obtain the final school state at time *t*. For the left figure, the fish 2 and 3 are ON because they are in the radius of centre of mass. For the left figure, the fish 1, 2, and 3 are ON because their direction (bold black arrow) are diverted from the average direction (bold red arrow). Subsequently, we compute Φ from a time series of these states using PyPhi. We assume that the network structure is similar to that of the local parameter settings, i.e., the fully connected network without self-loop.

**Figure 7 entropy-22-00726-f007:**
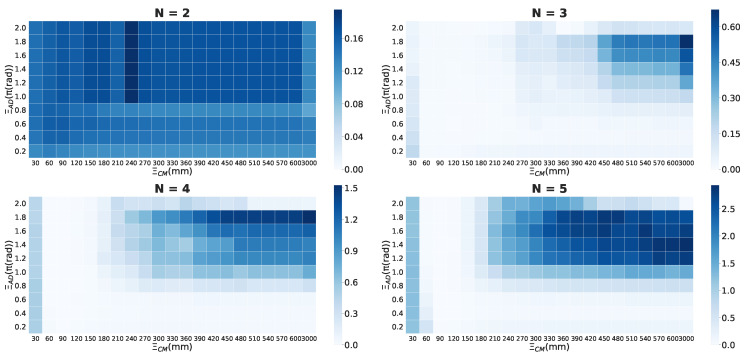
Heat maps on the global parameter settings for each group size (Δt=20/120 s). We took the average of 〈Φ(N)〉 for all datasets (colour bar). The horizontal axis shows the distance from the centre of mass ($\Xi_{CM}$), and the vertical axis shows the difference from average direction ($\Xi_{AD}$). The cells from $\Xi_{CM}=600$ to $\Xi_{CM}=3000$ were omitted owing to space limitations. All timescale figures are listed in Appendix A.

**Figure 8 entropy-22-00726-f008:**
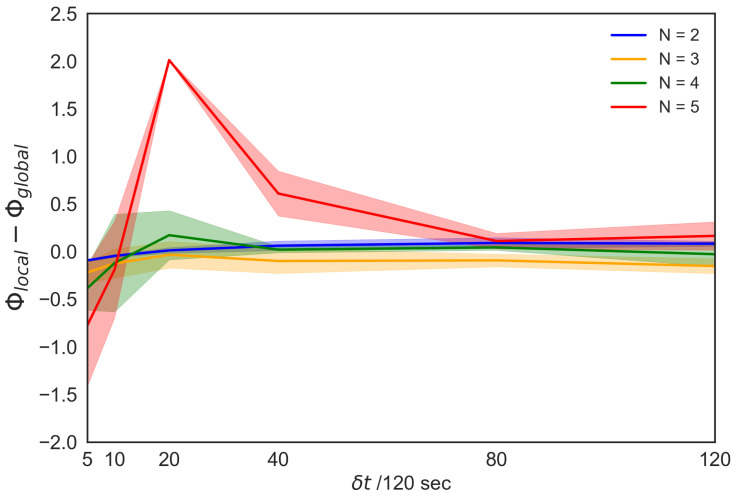
Difference between the average top 〈Φ(N)〉 value of the local and global group integrities. The horizontal axis shows the timescales (Δt=δt/120, where δt=5,10,20,40,80,120). The negative and positive values on the vertical axis represent that the global integrity over- or under-estimates the local integrity, respectively. The peak value of N=5 at Δt=20/120 is significantly greater than at the other peaks (Table 1).

**Figure 9 entropy-22-00726-f009:**
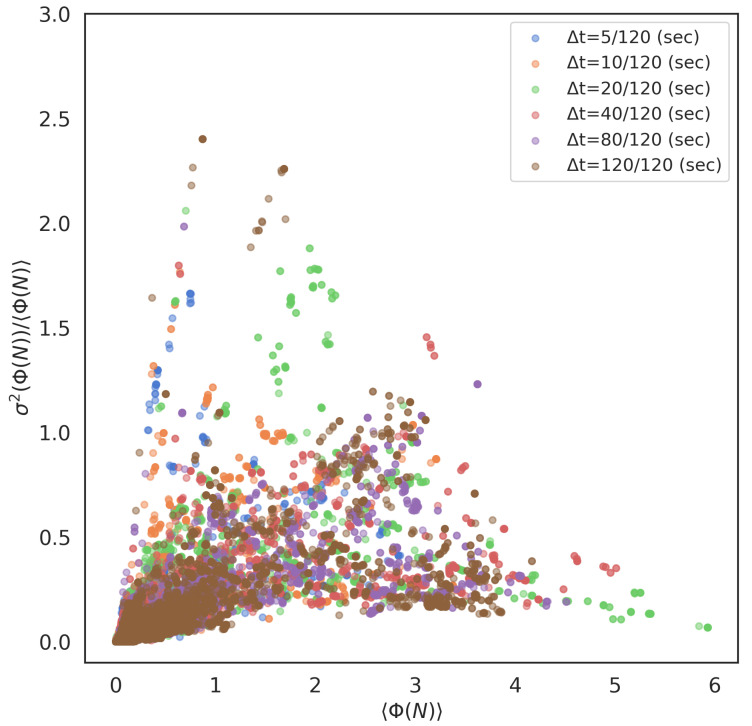
Correlation relation between 〈Φ(N)〉 and its Fano scales, $\sigma^{2}\left(\Phi \left(N\right)\right)/\langle \Phi \left(N\right)\rangle$ (normalized variance) for the local parameter setting. Each colour corresponds to timescales. The correlation coefficients are 0.66 (Δt=5/120 s), 0.71 (Δt=10/120 s), 0.61 (Δt=20/120 s), 0.72 (Δt=40/120 s), 0.75 (Δt=80/120 s), and 0.61 (Δt=120/120 s). For all the Pearson correlation tests, n=3200 (all data points for each scale: Appendix A) and $p<10^{-30}$. For the global parameter settings, see Appendix A.

**Figure 10 entropy-22-00726-f010:**
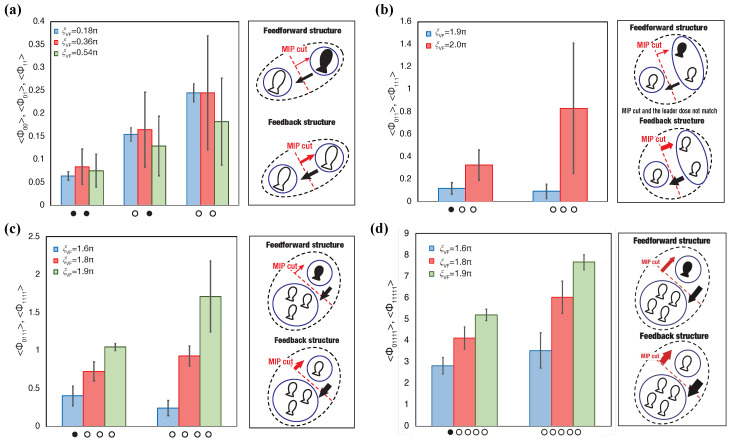
The classification from Φ values related to the school’s behaviour. (**a**) analysis of 2-fish schools. (**b**) analysis of 3-fish schools. (**c**) analysis of 4-fish schools. (**d**) analysis of 5-fish schools. We only compared the representative Φ values for each collective state from Table 2 parameter ranges. We fixed the turning rate parameter $\xi_{TR}=0.001$ (for other parameter settings, see Appendix A). We averaged Φ values for the same number of ON states (e.g., 01 and 10, etc.). For instance, over 99% collective states of 3, 4, and 5-fish’s school are all in the ON state (e.g., 111 for 3-fish schools) or the single OFF states (e.g., 011, 101, and 110 for 3-fish schools). We list all collective states for only the two-fish school because the rate of the collective states (11, 01, and 10) is approximately 20%∼50%. The horizontal circles represent the corresponding collective state in the same number of ON states (ON: white, OFF: black). The Φ values averaged the above-mentioned data over the distance parameters ($\xi_{D}=100$ for two-fish, $400\leq \xi_{D}\leq 1000$ for three-, four-, five-fish school) for each visual field parameter. The right box figure represents the information flow indicated from the left figure. The blue circle is the sub-group divided by the MIP cut (red dotted line). The red arrow is the cut flow and the black arrow is the opposite flow. The thickness of the arrow represents the intensity of the flow. From the MIP definition, the thickness of the black arrow is always greater than that of the red arrow. The statistical test is included in Appendix A.

**Table 1 entropy-22-00726-t001:** Tukey–Kramer method for Figure 8 at Δt=20/120 (s). The notation, $D_{G-L}\left(\Phi \left(N\right)\right)$, represents the difference between $\Phi_{local}\left(N\right)-\Phi_{global}\left(N\right)$ for group size *N*. We compute all pairs of the difference between different sizes at Δt=20/120 s for this statistical test.

	$D_{G-L}\left(\Phi \left(N\right)\right)-D_{G-L}\left(\Phi \left(M\right)\right)$	*p*-Value
N=2−M=3	−0.049	0.999
N=2−M=4	0.193	0.69
N=2−M=5	**2.31**	**<** $10^{-7}$
N=3−M=4	0.242	0.48
N=3−M=5	**2.36**	**<** $10^{-8}$
N=4−M=5	**2.12**	**<** $10^{-7}$

**Table 2 entropy-22-00726-t002:** The matching rate of the single OFF state individual and MIP cut or the positional leadership (PL). $\xi_{D}$ is set as follows. for N=2, $\xi_{D}=100$, for N=3,4,5, $400\leq \xi_{D}\leq 1000$. $\xi_{TR}$ is fixed at 0.001 (rad/step) because values are greater than other parameters (for $\xi_{TR}=0$ or 0.005, see Appendix A).

N	Visual Field $\xi_{VF}$, π(rad)	MIP Cut Match Rate (%)	PL Match Rate (%)
2	0.18	100	100
	0.36	100	100
	0.56	100	100
3	1.9	19	81
	2.0	26	40
4	1.6	18	92
	1.8	58	76
	1.9	67	42
5	1.6	70	91
	1.8	73	73
	1.9	100	42

**Table 3 entropy-22-00726-t003:** Data summary. These are all the data sets that we used in this paper. Three data sets for N=2,4,5 and four data sets for N=3. The total time length are approximately 10–15 min.

*N*	Average Distance (mm)	Average Velocity (mm/s)	Error (S.D.)	Minimum Distance (mm)	Cohesion Rate (%)	Total Time Steps
2	166.3	268.8	0.18	1.90	99.7 (99.9)	106,961
	90.67	271.68	0.23	0.10	99.8 (100)	99,431
	122.0	256.08	0.18	1.60	99.7 (100)	107,206
3	170.8	301.2	0.23	1.80	95.5 (97.9)	90,051
	159.1	343.2	0.14	1.83	98.8 (99.8)	83,654
	173.1	300.0	0.13	2.82	97.4 (99.0)	97,446
	132.0	240.0	0.19	1.67	99.2 (99.9)	93,931
4	164.3	270.72	0.14	1.18	98.8 (100)	106,327
	141.5	190.8	0.12	1.38	99.2 (100)	103,226
	114.9	148.56	0.38	1.83	99.8 (100)	98,126
5	143.8	259.92	0.28	0.79	98.7 (99.6)	102,895
	146.0	213.12	0.12	1.16	100 (100)	97,346
	143.7	259.2	0.28	1.44	97.2 (100)	92,116

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
