# Peer review of "Four-Types of IIT-Induced Group Integrity of Plecoglossus altivelis"

_entropy, 2020, doi:10.3390/e22070726_

Round 1

Reviewer 1 Report

The authors answered most of my concerns given in my previous report, and I applaud that the manuscript improved a lot. However, I still have one main concern. I list several minor points too.

Main concern:
The group size could have an effect on how the metrics are calculated. While also the behaviour of the groups with different number of individuals could be different. Differences based on group size are shown in the metrics, but at the moment, it is still not clear whether it is just an effect of the metrics themselves (if so, this could be seen as an artifact), or it is truly a difference in the behaviour. The authors now refer to their previous study, but as here they claim to observe new phenomena for example in case of N=5, it think in-depth analyses are needed, that sub-group of N=5 also show different characteristics then the real groups with that number of individuals.
One example to point out where such built in group size dependence can have an artificial effect, is in play in case the global parameter - distance from the centre of mass (CM). The authors argue that this is a value characterizing the clustering of the fish school. In a large, highly clustered, dense fish school where individuals are in close proximity to their neighbours, even if the local structure is unchanged for increasing group size, the parameter used in this analyses will change drastically. And the average distance from the CM will increase, and as the ON/OFF states are calculated based on a cut-off distance, some individuals actually connected to the cluster will still got an OFF state by the analysis.

Minor comments:

Title: please write out "IIT"

Abstract: Line 8-9: "Second, we set several timescales ..." This is stated, but later in the abstract there is no connection why this is important, and what effect it has on the findings.

Line 63-66: "Indeed, IIT-induced leadership is never replicated by other measures (i.e., mutual information and the sum of the transfer entropy) and under some artificial conditions (Boid and homogeneous Markov models). In other words, the discontinuity between three and four schools of fish in IIT 3.0 was never observed on the mutual information and the sum of the transfer entropy."
This is a point I already raised in previous review, and I am not satisfied with the version presented. The phrase "In other words" connects the two sentences, but the meaning of the two sentences are very different. Also the first sentence is a crucial statement. If the first sentence is true, that there is a need for a justification that the "IIT-induced leadership" is a meaningful concept, and actually relates to leadership. The authors later write about that they performed some analyses, so that should be clearly connected here. And this part should be rephrased.

Line 97: Equations 7-8: it is not defined what is D and what is the meaning of ||.

Fig. 1a: the 3 panels should show the same moment in time for a group of 4 fish. However the layout is slightly different in all of the 3 panels. This maybe a really minor comment, but I think this gives a bit of an impression of being careless which I am sure is not what characterizes the author's work.

Fig. 2: axis label "Visual Feild" -> "Visual Field"

Fig. 4a: the y axis is incorrect

Line 328 and Fig. 9 legend (multiple times): "corrective" -> "collective". The safest would be to check everywhere

Summary:

As a consequence of all above, I still can't support acceptance.

Author Response

Responses to Reviewer 1:

Thank you again for taking the time to review our manuscript. We have addressed all of your comments, as described in the dialogue below.

Major concerns:

The group size could have an effect on how the metrics are calculated. While also the behaviour of the groups with different number of individuals could be different. Differences based on group size are shown in the metrics, but at the moment, it is still not clear whether it is just an effect of the metrics themselves (if so, this could be seen as an artifact), or it is truly a difference in the behaviour. The authors now refer to their previous study, but as here they claim to observe new phenomena for example in case of N=5, it think in-depth analyses are needed, that sub-group of N=5 also show different characteristics then the real groups with that number of individuals.

One example to point out where such built in group size dependence can have an artificial effect, is in play in case the global parameter - distance from the centre of mass (CM). The authors argue that this is a value characterizing the clustering of the fish school. In a large, highly clustered, dense fish school where individuals are in close proximity to their neighbours, even if the local structure is unchanged for increasing group size, the parameter used in this analyses will change drastically. And the average distance from the CM will increase, and as the ON/OFF states are calculated based on a cut-off distance, some individuals actually connected to the cluster will still got an OFF state by the analysis.

  • First, we apologies that our calculation was a mistake on five-fish school for Figure 9 (10). We found that four- and five-fish school’s causal structures (feed-back and feed-forward) are the same. Therefore, the analysis of our previous study become sufficient to indicate the difference for two- and three-fish school. The revised edition includes more detail descriptions on them (L536-556).
  • Although we could not find the difference of causal structure in four- and five-fish school, we found that the Φ values are different at some parameter settings. In the revised manuscript, while pointing out the above concerns, this study argues that the N=4 and N=5 systems are distinguishable in terms of the global and the local integrities. Specifically, we argue that the global interaction can be interpreted as the mean-field approximation and that the difference between the global and local integrity as the degree of heterogeneous interaction (L383-L402). In the long timescales, this interaction heterogeneity diminishes. This result proves the approximation also work at the group integrity. We then discussed that around the response time of the ayus (Figure S7, newly added), homogeneous interactions underestimate the local interactions at N=5. We believe that this allows us to claim the difference between N=4 and 5 with some clarity (L591-L618).

Minor comments:

Title: please write out "IIT"

  • The point has been fixed.

Abstract: Line 8-9: "Second, we set several timescales ..." This is stated, but later in the abstract there is no connection why this is important, and what effect it has on the findings.

  • We added the description on the time scale analysis (L9-L12).

Line 63-66: "Indeed, IIT-induced leadership is never replicated by other measures (i.e., mutual information and the sum of the transfer entropy) and under some artificial conditions (Boid and homogeneous Markov models). In other words, the discontinuity between three and four schools of fish in IIT 3.0 was never observed on the mutual information and the sum of the transfer entropy."

This is a point I already raised in previous review, and I am not satisfied with the version presented. The phrase "In other words" connects the two sentences, but the meaning of the two sentences are very different. Also the first sentence is a crucial statement. If the first sentence is true, that there is a need for a justification that the "IIT-induced leadership" is a meaningful concept, and actually relates to leadership. The authors later write about that they performed some analyses, so that should be clearly connected here. And this part should be rephrased.

  • As the reviewer points out, this part of the description was unclear, so we have re-written it to separate the discontinuity descriptions from the leadership descriptions (L64-L77).

Line 97: Equations 7-8: it is not defined what is D and what is the meaning of ||.

  • The point has been fixed.

Fig. 1a: the 3 panels should show the same moment in time for a group of 4 fish. However the layout is slightly different in all of the 3 panels. This maybe a really minor comment, but I think this gives a bit of an impression of being careless which I am sure is not what characterizes the author's work.

  • In the revised edition, we changed Figure 1 to be able to understand all depicted situations are at the same moment.

Fig. 2: axis label "Visual Feild" -> "Visual Field"

Fig. 4a: the y axis is incorrect

Line 328 and Fig. 9 legend (multiple times): "corrective" -> "collective". The safest would be to check everywhere

  • These points have been fixed.

We would like to sincerely appreciate your insightful and constructive comments and suggestions. We believe that these have greatly strengthened the paper.

Thank you once again for taking the time to review our manuscript.

Yours sincerely,

Takayuki Niizato

Department of Intelligent Interaction Technologies

University of Tsukuba

Tennodai 1-1-1, Tsukuba,

Ibaraki, Japan 305-8577

Reviewer 2 Report

The quality and clarity of the article has been greatly improved. Some of the improvements listed below may further enhance this paper before publication.

Nevertheless, the information provided on data quality identified a major weakness in this publication. Only 3 replications were carried out by group size. As inter-group variability is important in social fish (see Collignon (2019) Collective departures and leadership in zebrafish), this low number of replicates weakens the results obtained. More problematic, Table 3 shows a cohesion rate of 1% instead of the >95% obtained for the rest of the experiments. This data potentially invalidates the results obtained on group sizes of 5 fish. It is imperative that the authors justify this information.

Minor comments and suggestions:

- Figure 1 could be separated into two independent figures to make the legend more readable.

- P13, l361: The authors indicate that their results correspond to the reaction time of the fish Hemigrammus rhodostomus. As the two fish species have very little in common (the hemigrammus are 10 times smaller for example), I do not understand the relevance of this comparison?

- Several times in this article, the authors address the readers directly (l186 or 518 for example). This seems a bit out of tone in this paper and could be better rephrased.

- The discussion lacks a real conclusion on the results obtained on the different time scales.

Author Response

Responses to Reviewer 2:

Thank you again for taking the time to review our manuscript. We have addressed all of your comments, as described in the dialogue below.

Nevertheless, the information provided on data quality identified a major weakness in this publication. Only 3 replications were carried out by group size. As inter-group variability is important in social fish (see Collignon (2019) Collective departures and leadership in zebrafish), this low number of replicates weakens the results obtained. More problematic, Table 3 shows a cohesion rate of 1% instead of the >95% obtained for the rest of the experiments. This data potentially invalidates the results obtained on group sizes of 5 fish. It is imperative that the authors justify this information.

  • That was typo (1.00% -> 100%). In the revised paper, we also added the data when the threshold is 700(m) (L707-711).

Minor comments and suggestions:

Figure 1 could be separated into two independent figures to make the legend more readable.

  • We divided these two figures.

- P13, l361: The authors indicate that their results correspond to the reaction time of the fish Hemigrammus rhodostomus. As the two fish species have very little in common (the hemigrammus are 10 times smaller for example), I do not understand the relevance of this comparison?

  • In this revised edition, we calculated the directional correlation function (Nagy et al, 2010) for the reaction time of ayus. Figure S7 shows its results. The peak of the graph locates between 0.2 and 0.4 (sec) (L762).

Several times in this article, the authors address the readers directly (l186 or 518 for example). This seems a bit out of tone in this paper and could be better rephrased.

  • We have rephrased sentences that the reviewer feels unnatural.

The discussion lacks a real conclusion on the results obtained on the different time scales.

  • We added the timescale discussion related to the mean-field approximation. The heterogeneous interaction gradually shifts to the homogeneous one along with the timescales (L603-L618).

We would like to sincerely appreciate your insightful and constructive comments and suggestions. We believe that these have greatly strengthened the paper.

Thank you once again for taking the time to review our manuscript.

Yours sincerely,

Takayuki Niizato

Department of Intelligent Interaction Technologies

University of Tsukuba

Tennodai 1-1-1, Tsukuba,

Ibaraki, Japan 305-8577

Reviewer 3 Report

My fundamental issue remains unaddressed: The data being fed into IIT is in a form that is very difficult to interpret. Why would the "active" state as defined by the authors be expected to be predictive? (If some set of individuals are "active" now, do I really expect to be able to predict which individuals will be "active" next?)  How do we know that the results are not an artifact of how the authors set up the coarse-graining?

The authors state in their response that they chose the coarse-graining "so that we can immediately see in the model whether our analysis is just an artificial result". This is not clear to me---what checks have you done to know that it's not an "artificial result"?

The authors attempt to justify the coarse-graining by claiming that it relates closely to the BOID model. First, I think calling the BOID model the "most well-applicable model" to biology is a stretch. The BOID model was mostly used for computer animation, and many other models of animal group motion have been created since then that are more closely tied to real data. Second, and more to the point, the coarse-graining implemented by the authors is not standard in any model of animal behavior that I am aware of, and therefore needs to be justified.

Also note that I am not asking the authors to avoid coarse-graining altogether (to, for instance, use continuous quantities). There are other ways to coarse-grain the data that would not, for instance, lose information about whether fish are inactive due to being far from one another or because they are not actively turning.

I am not sure how Figure 9 addresses the "locality" issue that I brought up in my previous review. Table 2 seems to show that a single OFF individual is sometimes, but not always, the "positional leader" (except for 2 fish schools, where this always holds, maybe by definition). How does this connect with my concern that the analysis does not distinguish cases where, e.g., the OFF individual is near or far from other fish?

I also remain concerned that the authors are over-interpreting Phi by assuming that parameters for which it is maximized can be used to define effective values for those parameters. As far as I am aware, this is not standard in the field. The authors state in their rebuttal letter "the analysis of maximal values for certain parameters can be found in other studies besides this one, such as the analysis of transfer entropy (see Introduction)", but I am unable to find where the authors discuss previous results in this context or make any argument for their particular use of maximizing Phi.

The distinction between "internal" and "external" is still confusing to me, but this issue is not as important as the above issues.

The authors did not respond convincingly to my question about comparing Phi in "local" and "global" contexts (Figure 7). In the rebuttal, they point to the five fish school results in Figure 9, but I cannot determine how this is related to "local" versus "global" (or "top-down" versus "bottom-up").

Finally, I reassert that more needs to be explained about Table 1. If the authors are calculating a p-value, then they must be assuming some distribution over Phi values. Figure 7 appears to be showing ranges of Phi values, which may represent the widths of such distributions, but this is not specified. On line 346, we see "Figure 7 shows the comparison between the mean <Phi(N)> values (top 20 for each)". Perhaps the authors are showing in Figure 7 the range of the top 20 values as parameters are changed? If this is the case, and if this is what is being used to construct Table 1, then the calculated p-values do not make much sense to me. This would be incorrectly conflating Phi calculated at different parameter values with independent samples from some distribution.

I strongly believe that this manuscript is still not ready for publication.

A final additional question:

I'm unsure about the interpretation of Figure 9.  How is a Phi value calculated for a particular state (e.g. the all ON state in Figure 9)? Doesn't Phi depend on the relationships between all observed states? Are you instead computing the particular state's contribution to Phi?

Author Response

Responses to Reviewer 3:

Thank you again for taking the time to review our manuscript. We have addressed all of your comments, as described in the dialogue below.

My fundamental issue remains unaddressed: The data being fed into IIT is in a form that is very difficult to interpret. Why would the "active" state as defined by the authors be expected to be predictive? (If some set of individuals are "active" now, do I really expect to be able to predict which individuals will be "active" next?)  How do we know that the results are not an artifact of how the authors set up the coarse-graining?

  • First, IIT 3.0 measures the predictability from collective state to another collective state, not inter-individual predictabilities. More precisely, IIT 3.0 describes the inter-subgroup interactions in the given system. Therefore, we are not investigating the OFF state individual's contribution to the next individual states. Rather, we are investigating that the state of the whole system, including the OFF states, affects the next state of the whole system.
  • Comparisons with "artificial" results have been discussed in previous studies (Niizato et al., 2020) through comparisons with the BOID model and with random Markov processes. Through these comparisons, we discussed whether the effect of the coarse-graining or the IIT 3.0 itself was the matter. In the revised edition, we added detail comments on the previous study (L64-L77).

The authors state in their response that they chose the coarse-graining "so that we can immediately see in the model whether our analysis is just an artificial result". This is not clear to me---what checks have you done to know that it's not an "artificial result"?

  • (As I have answered the question above) A comparison to the BOID model with real data measures the real effect of actual school movements in IIT.

The authors attempt to justify the coarse-graining by claiming that it relates closely to the BOID model. First, I think calling the BOID model the "most well-applicable model" to biology is a stretch. The BOID model was mostly used for computer animation, and many other models of animal group motion have been created since then that are more closely tied to real data. Second, and more to the point, the coarse-graining implemented by the authors is not standard in any model of animal behavior that I am aware of, and therefore needs to be justified.

Also note that I am not asking the authors to avoid coarse-graining altogether (to, for instance, use continuous quantities). There are other ways to coarse-grain the data that would not, for instance, lose information about whether fish are inactive due to being far from one another or because they are not actively turning.

  • First of all, we do not think that BOID is used mainly in computer simulations. Many studies in collective behaviour assume that the BOID framework (neighbourhoods and direction changes) is common, despite their differences. However, as the reviewer had suggested, the coarse-graining of BOID is not a common assumption, and so is needed to justify.
  • Therefore, we have added a new description of this ambiguity from the coarse-graining (out of the visual field or non-active turning) in the revised edition (L231-L250). The ambiguity of the binary is not limited to animal groups but is even found in biological networks (we cannot discriminate the OFF state as inhibited or no-input). Therefore, the critical point for the analysis is the correlation between internal causal structures and external behaviours. We discuss that, in the revised manuscript, while pointing out the ambiguity that the reviewer had suggested, the relationship with external behaviours must become one-to-many relations rather than one-to-one relations. Our analysis would give some hints to the researchers who try to understand human consciousness by using IIT 3.0 because they finally end up to consider this one-to-many relation between the brain states and human behaviour.

I am not sure how Figure 9 addresses the "locality" issue that I brought up in my previous review. Table 2 seems to show that a single OFF individual is sometimes, but not always, the "positional leader" (except for 2 fish schools, where this always holds, maybe by definition). How does this connect with my concern that the analysis does not distinguish cases where, e.g., the OFF individual is near or far from other fish?

  • The case pointed out rarely happens because the cohesion rate is very high for the parameter settings in Figure 9 (10) (L707-L711). In other words, the OFF state means that other individuals are either out of eyeshot or non-turning direction. Too far distances result in low Φ values. The description of how the two kinds of OFF states are related to the external behaviour is newly added to the revised edition (L536-L556).

I also remain concerned that the authors are over-interpreting Phi by assuming that parameters for which it is maximized can be used to define effective values for those parameters. As far as I am aware, this is not standard in the field. The authors state in their rebuttal letter "the analysis of maximal values for certain parameters can be found in other studies besides this one, such as the analysis of transfer entropy (see Introduction)", but I am unable to find where the authors discuss previous results in this context or make any argument for their particular use of maximizing Phi.

  • We added three reasons why we chose the maximum Φ values (L257-L268).
  • (i) Some studies apply the maximum information to select the appropriate time scale for the analysis (Crosato et al., 2017 in Figure 7 and Engel et al., 2018).
  • (ii) High Φ values in IIT 3.0 mean that the system has an abundance of feedback structure. This feedback structure has been treated as one of the essential indicators in system theory (Bertalanffy, 1969 and Goldberg, 1969).
  • (iii) The relationship between IIT and critical phenomena has been pointed out in several studies (Aguilea, 2019, Sheneman et al., 2019, Nilsen et al., 2019). In addition, we also observed the critical phenomena in the movement of ayus (Murakami and Niizato, 2017)

The distinction between "internal" and "external" is still confusing to me, but this issue is not as important as the above issues.

  • As already mentioned, Φ in IIT3.0 is derived from the transition networks of each collective state. In other words, instead of examining the information flows between individuals, we focus on the information flow in the process of the whole system. Therefore, the internal causal process need not correlate with the external behaviour (the MIP-cut never always correlates with the leadership in Figure 9 (10)).

The authors did not respond convincingly to my question about comparing Phi in "local" and "global" contexts (Figure 7). In the rebuttal, they point to the five fish school results in Figure 9, but I cannot determine how this is related to "local" versus "global" (or "top-down" versus "bottom-up").

  • In the revised edition, we added the description which insists that the computation of the global integrity shares the same framework on the mean-field approximation in section 2.3 (L383-L402). The global group integrity computes the group integrity with assuming the homogeneous interaction. This kind of assumption works in long time scales (the difference between global and local integrity approaches to zero). In the relatively short timescale, the difference between them indicates the degree of the local integrity can under- or over-estimate with assuming the homogeneous interaction.

Finally, I reassert that more needs to be explained about Table 1. If the authors are calculating a p-value, then they must be assuming some distribution over Phi values. Figure 7 appears to be showing ranges of Phi values, which may represent the widths of such distributions, but this is not specified. On line 346, we see "Figure 7 shows the comparison between the mean <Phi(N)> values (top 20 for each)". Perhaps the authors are showing in Figure 7 the range of the top 20 values as parameters are changed? If this is the case, and if this is what is being used to construct Table 1, then the calculated p-values do not make much sense to me. This would be incorrectly conflating Phi calculated at different parameter values with independent samples from some distribution.

  • As the reviewer had pointed out, there are some ambiguous descriptions on selecting the Phi values. So, we added some descriptions in Section 2.3 (L403-L416). As we mentioned above, the difference between global and local group integrity measures the degree of heterogeneity of the local interaction. Although each Phi values are picked up from the different parameter settings, our interest is to measure that “the ability of the system to generate a causal power within different parameter settings” compared with assuming the homogeneous interaction. Therefore, we do not confuse here that the Phi calculated at different parameter with independent samples from some distribution.

A final additional question:

I'm unsure about the interpretation of Figure 9.  How is a Phi value calculated for a particular state (e.g. the all ON state in Figure 9)? Doesn't Phi depend on the relationships between all observed states? Are you instead computing the particular state's contribution to Phi?

  • Although IIT 3.0 can only handle discrete systems, the discrete system enables us to consider Φ for each collective state (e.g. 00, 01, 01, 11) (see L251-255). This discreteness enables us to compare the IIT 3.0 analysis with the actual collective behaviour. So, we have the Φ00, Φ01, Φ10, Φ11. Of course, as the reviewer had mentioned (“Phi depend on the relationships between all observed states”), all possible past and future collective states are involved to determine the Φ values; Φs can be computed for all collective states (that’s the default for PyPhi computation).

We would like to sincerely appreciate your insightful and constructive comments and suggestions. We believe that these have greatly strengthened the paper.

Thank you once again for taking the time to review our manuscript.

Yours sincerely,

Takayuki Niizato

Department of Intelligent Interaction Technologies

University of Tsukuba

Tennodai 1-1-1, Tsukuba,

Ibaraki, Japan 305-8577

Round 2

Reviewer 2 Report

The previous remarks have been well integrated.
However, the authors did not respond to my concerns about the weakness of the data set regarding the low number of replications by group size.

"Nevertheless, the information provided on data quality identified a major weakness in this publication. Only 3 replications were carried out by group size. As inter-group variability is important in social fish (see Collignon (2019) Collective departures and leadership in zebrafish), this low number of replicates weakens the results obtained."

Can you justify the reason for this design?

Author Response

Responses to Reviewer 2:

Thank you again for taking the time to review our manuscript. We have addressed all of your comments, as described in the dialogue below.

The previous remarks have been well integrated. However, the authors did not respond to my concerns about the weakness of the data set regarding the low number of replications by group size. "Nevertheless, the information provided on data quality identified a major weakness in this publication. Only 3 replications were carried out by group size. As inter-group variability is important in social fish (see Collignon (2019) Collective departures and leadership in zebrafish), this low number of replicates weakens the results obtained." Can you justify the reason for this design?

  • The inter-group variability, as Collignon et al. had suggested, also exists in ayu; as our analysis had shown, the variances of Φs within the same group size were not so large (in Figure 10). Besides, our previous studies showed that the switching time of positional leadership obeys the power-law distribution for all datasets. In this sense, the statistical properties of our dataset can regard the approximately same in terms of one system. Hence, we consider the inter-group variability in this study never affects our results critically.
  • We added this description in Section 4. 4.

We would like to sincerely appreciate your insightful and constructive comments and suggestions. We believe that these have greatly strengthened the paper.

Thank you once again for taking the time to review our manuscript.

Yours sincerely,

Takayuki Niizato

Department of Intelligent Interaction Technologies

University of Tsukuba

Tennodai 1-1-1, Tsukuba,

Ibaraki, Japan 305-8577

This manuscript is a resubmission of an earlier submission. The following is a list of the peer review reports and author responses from that submission.

Round 1

Reviewer 1 Report

In the manuscript titled “New Classification of Collective Animal Behaviour as an Autonomous System” the authors report a collective behaviour study based on experiments with groups consisted of 2 to 5 fish (ayus, Plecoglossus altivelis) swimming freely in a large tank (3x3m2). They propose an information theory-based approach to analyse the data, and find quantitative differences for the different group sizes. They analyse the data for a large range of parameter settings as well as on multiple temporal scale. They interpret the differences found based on the quantitative output of the used analysis method. I find the concept useful and interesting, also there are several thoughts in the manuscript that could provide a new direction and a step up for collective behaviour research. The distinction between top-down and bottom-up level quantification of the collective behaviour is one of the great ideas of the manuscript and the systematic comparison between the two could serve valuable insight. That said, I have several major problems with the manuscript in its current form. I give a detailed list below. I also have some minor comments or suggestions for correction.

Major concerns:

- The manuscript in its current form is not a standalone paper. It refers in several points to a previous paper of the authors (Ref. 77 of the manuscript). Also it uses terminology that requires in-depth knowledge of the “integrated information theory”. I think the integrity of the paper should be improved.

- A crucial point is that at the moment it is not clear where do the observed differences in the calculated metrics for different group sizes come from. Is this an inherent property of the system and the difference in the metric is a meaningful description of the behaviour of the group, or rather it is an artificial product of how the metric is calculated for different N.

One suggestion to show that there is clear difference between the group-level behaviour, would be to define subset from a larger group(s) and calculate the metrics using these data. So for example, a randomly selected 2, 3 or 4 individual from the group of 5, analysed separately as if they would belong to a group of 2, 3 or 4 would show the similar or different metrics as the real group of 2, 3 or 4 (respectively)? If the metrics are similar then the difference is rather coming from how the calculations are made, and not a real characteristic of the system itself. I am not implying that this is the case, but at the moment, there is no way for the reader to decide.

- Currently, only aggregated metrics are derived and shown, and it is hard to relate the actual behaviour of the fish to these. The discussion provided some interpretation by the authors, but this is not based on evidence (at least not shown). How does these cases look for the different regimes (high or low values of Phi? Please show raw data - trajectories of the fish: 1. filter from the real data set or 2. generated trajectory data that results in similar metrics as the different experimental cases - to support your interpretation.

- Somewhat related to the previous 2 points, is the question whether a possible tautology is happening here? The authors (re)define terminology commonly used in the collective behaviour literature in case of, for example, "leadership", "followership", “fission-fusion”, “Boid-like”, etc., but no real insight is given other than stated. Maybe their new definitions captures well the original phenomena that are typically being used in the research field. But without further support and direct justification (see the previous point), it looks like the authors define new terms, and show how the systems with different group size behave like the new definition.   

- It is not clear how many experimental trails were performed. Does Table 2 contains information from all of them? Please state explicitly. Also how representative the measured metric is? If randomly splitting the datasets, could the outcome of the metric robustly reproduced?

Minor comments and suggestions:

- Title: should be much more specific, should contain information on the method and the specific study system

- Abstract and later in the main text: There is not clear what is the difference between “IIT” and “IIT 3.0”. Also this should be made clear and should be used consistently

- Abstract: I am not convinced by the interpretation of the observed differences in the metrics (as described in the major concerns), so the sentences starting from “The concrete classification includes…” are yet not justified in the later text.

- Line 13: "These minute classifications": I don't understand

- Line 66: “IIT-induced leadership is never replicated using other measures (i.e., mutual information and the sum of the transfer entropy) and under some artificial conditions (Boid and homogeneous Markov models).” This statement underlines the need for a convincing and in-depth justification as raised in the major concerns.

- Line 81: define X earlier

- Line 83: TPM used before it is defined in Line 168

- the local parameter settings consist of distance, visual field and turning rate. In the text this is justified by previous work (like Boids or other zonal models). Still here two qualitatively different things are treated the same way. The field of view defines the “input” a fish receives. Having the On/Off binary state for it makes sense. The turning rate is rather a behavioural "output". The distance could be seen either ways, as this could determine whether the fish can get influenced by a neighbour (input), or as a behavioural output that results in schooling. In the text the authors relate the turning rate to the (angular) noise used typically in simulations. I find this rather confusing and misleading.

In some cases the fish can be in “All-ON” state for example, even if there is only a faraway individual within the field of view but outside the distance threshold, while another individual in its blind spot but within the distance threshold. Is this - and other similar examples – “outlier”, an extreme case of something happening very rare for the different system sizes, or rather a typical scenario.

- Fig. 1: The figure is misleading and/or ambiguous. Two possible scenarios that, 1) Any parts of the neighbouring fish intersecting with the circle that defines the zone of this interaction, would mean that their stage is ON. 2) The center (centre of mass?) of the neighbouring fish needs to be within the circle for the ON stage. Or is there any third option? Please make it clear. If using criteria 1 then on the left panel Fish_4 is in the zone of Fish_2 and on the middle panel Fish_2 is within the visual field of Fish_1. Criteria 2 also not consistent with the figure.

The visual field is presented by green, and not blue.

- Fig. 2: On each panel the scale (size of balls and the colour scale) of the visualization is different. How does this allow the comparison of the results? This relates to the second major point about what comes directly from the number of fish in the statistics not related to their behaviour.

- Line 177: “see 4” is this the Ref. 4?

- Line 195: “values represent the group’s integrity in the school (of fish)” this needs justification

- Fig. 3a legend title: write 0.05, instead of .05

- Line 226-228: Both (i) and (ii) should be rephrased. It is not clear at the moment.

- Fig. 4a: what is the meaning of the brown square at (0,0). That was supposed to represent N=4 and N=5 all timescales on top of each other? If there is no data in that range, then is this why it is defined as 0?

- Line 245: Normalizing with the observed maximum value: this can have a crucial effect on the numbers, if a fully connected state occurs rarely. That would mean that an occasionally high value rescales the metric.

- Fig. 6: the last bin of the x axis on all 4 panels states 3000. Is this correct? If so, still should be explained.

- Fig. 7: How does using only the top values affects this result?

- Fig. 8: What n were used in the Pearson correlation tests? All data points? Are those data point statistically independent? Please explain/justify.

- Discussion: I do not understand why the “original heap paradox” is discussed in details, rather its consequences/implications to group behaviour

- Line 365-367: this is an example of the possible tautology raised in the major concerns

- Table 2: the table legend is not very useful, as basically just repeats the title row of the table. Also what is the reason writing “(mm) per second”? Proper description here would help the reader

Note:

I could not access the Supplementary Figures.

Summary:

As a consequence of all above, I suggest major revision. I would be willing to review a much improved version.

Author Response

Responses to Reviewer 1:

Thank you again for taking the time to review our manuscript. We have addressed all of your comments, as described in the dialogue below.

Major concerns:

The manuscript in its current form is not a standalone paper. It refers in several points to a previous paper of the authors (Ref. 77 of the manuscript). Also it uses terminology that requires in-depth knowledge of the “integrated information theory”. I think the integrity of the paper should be improved.

  • We added a review of the previous research in the Introduction section. At the end of section 2.1, a description of the IIT 3.0 terms used in this paper and the correspondence between their biological meanings and the IIT terminology is given.
  • Especially, in the revised edition, new numerical analysis has been added (new Figure 9 and new Table 2) to reduce the reliance on past research as much as possible, and the paper can be read as independent.

A crucial point is that at the moment it is not clear where do the observed differences in the calculated metrics for different group sizes come from. Is this an inherent property of the system and the difference in the metric is a meaningful description of the behaviour of the group, or rather it is an artificial product of how the metric is calculated for different N.

  • We hope that the following statements makes answers.

One suggestion to show that there is clear difference between the group-level behaviour, would be to define subset from a larger group(s) and calculate the metrics using these data. So for example, a randomly selected 2, 3 or 4 individual from the group of 5, analysed separately as if they would belong to a group of 2, 3 or 4 would show the similar or different metrics as the real group of 2, 3 or 4 (respectively)? If the metrics are similar then the difference is rather coming from how the calculations are made, and not a real characteristic of the system itself. I am not implying that this is the case, but at the moment, there is no way for the reader to decide.

  • Comparison of subsystems for all schools (e.g., 2, 3, and 4-fish in five fish) confirmed the differences between the full size of the schools. For example, a followership-like distribution (in this revised version, "chasing") could not be identified for any of the other 3, 4, and 5 2-subsystems (Niizato et al., 2019). Hence, our results are not derived from the method of calculation itself, but from its causal structure in the collective behaviour. A detailed description of this has been added to the Discussion.

Currently, only aggregated metrics are derived and shown, and it is hard to relate the actual behaviour of the fish to these. The discussion provided some interpretation by the authors, but this is not based on evidence (at least not shown). How does these cases look for the different regimes (high or low values of Phi? Please show raw data - trajectories of the fish: 1. filter from the real data set or 2. generated trajectory data that results in similar metrics as the different experimental cases - to support your interpretation.

  • The revised manuscript includes comparisons between the integrated information Φ and specific collective behaviors. First, we added a new Table 2 of positional/information-related correspondences to the actual collective behaviour. Next, we added new Figure 9 with the relationship between Φ value and MIP-cut corresponding to the primal (99% of the events) corrective states in subsystem. We also added an example of actual positional relationships for some exceptional behaviour (Figure S7).

Somewhat related to the previous 2 points, is the question whether a possible tautology is happening here? The authors (re)define terminology commonly used in the collective behaviour literature in case of, for example, "leadership", "followership", “fission-fusion”, “Boid-like”, etc., but no real insight is given other than stated. Maybe their new definitions captures well the original phenomena that are typically being used in the research field. But without further support and direct justification (see the previous point), it looks like the authors define new terms, and show how the systems with different group size behave like the new definition.  

  • In the revised manuscript, we reveal numerical differences from the newly added Table 2 and Figure 9 mentioned earlier, based on the integrated information Φ and the location of MIP, for systems ranging from two- to five-fish schools. In particular, a detailed analysis of the five-fish revealed new differences that were not seen in the previous study. This result is one example of new facts that have been revealed by taking into account long time scales.

It is not clear how many experimental trails were performed. Does Table 2 contains information from all of them? Please state explicitly. Also how representative the measured metric is? If randomly splitting the datasets, could the outcome of the metric robustly reproduced?

  • Table 2 (or Table 3 in the revised version) contains all the information. All of this data was used. In addition, a brief description of the experimental data (number of experiments and total time) was added at the beginning in Section 2.2.

Minor comments and suggestions:

Title: should be much more specific, should contain information on the method and the specific study system

  • The title has been changed in the revised version.

Abstract and later in the main text: There is not clear what is the difference between “IIT” and “IIT 3.0”. Also this should be made clear and should be used consistently

  • Except for the first sentence, IIT has been revised so that the reader can understand as IIT 3.0.

Abstract: I am not convinced by the interpretation of the observed differences in the metrics (as described in the major concerns), so the sentences starting from “The concrete classification includes…” are yet not justified in the later text.

  • The revised version includes new descriptions, figures, and tables as described above. We hope these are persuasive in this revised version.

Line 13: "These minute classifications": I don't understand

  • The word “minute” was eliminated.

Line 66: “IIT-induced leadership is never replicated using other measures (i.e., mutual information and the sum of the transfer entropy) and under some artificial conditions (Boid and homogeneous Markov models).” This statement underlines the need for a convincing and in-depth justification as raised in the major concerns.

  • This conclusion is the one of our previous study. In the revised section, we have included a longer explanation of this part.

Line 81: define X earlier

  • Changed to give the definition first.

Line 83: TPM used before it is defined in Line 168

  • The point that the reviewer mensioned have been corrected.

the local parameter settings consist of distance, visual field and turning rate. In the text this is justified by previous work (like Boids or other zonal models). Still here two qualitatively different things are treated the same way. The field of view defines the “input” a fish receives. Having the On/Off binary state for it makes sense. The turning rate is rather a behavioural "output". The distance could be seen either ways, as this could determine whether the fish can get influenced by a neighbour (input), or as a behavioural output that results in schooling. In the text the authors relate the turning rate to the (angular) noise used typically in simulations. I find this rather confusing and misleading.

  • Certainly the description associating angle changes with noise, as the reviewer says, is inappropriate. The point has been corrected.

In some cases the fish can be in “All-ON” state for example, even if there is only a faraway individual within the field of view but outside the distance threshold, while another individual in its blind spot but within the distance threshold. Is this - and other similar examples – “outlier”, an extreme case of something happening very rare for the different system sizes, or rather a typical scenario.

  • We counted the presence of individuals within 40 cm throughout all steps and found that all of them were above 95% of the time (defined cohesion rate in Table 3). This means that such events that the reviewer points out are rare in our data.

Fig. 1: The figure is misleading and/or ambiguous. Two possible scenarios that, 1) Any parts of the neighbouring fish intersecting with the circle that defines the zone of this interaction, would mean that their stage is ON. 2) The center (centre of mass?) of the neighbouring fish needs to be within the circle for the ON stage. Or is there any third option? Please make it clear. If using criteria 1 then on the left panel Fish_4 is in the zone of Fish_2 and on the middle panel Fish_2 is within the visual field of Fish_1. Criteria 2 also not consistent with the figure.

  • The former reviewer's interpretation is correct. The problem was that the picture of the fish was too big. To avoid such misunderstandings, we have redrawn the figures and added a detailed explanation of each example to the caption in Figure 2.

The visual field is presented by green, and not blue.

  • The point has been fixed.

Fig. 2: On each panel the scale (size of balls and the colour scale) of the visualization is different. How does this allow the comparison of the results? This relates to the second major point about what comes directly from the number of fish in the statistics not related to their behaviour.

  • Here the visualization is scaled only to support the reader's intuitive understandings. The scale problem does not show up in the actual analysis because we are comparing the normalized Φ values; the comparison between Φ values and the actual behaviour of individuals is a new addition to the revised manuscript (Figure 9 and Table 2).

Line 177: “see 4” is this the Ref. 4?

  • That was a typo. We meant to indicate Table 2(Table 3 in revised version).

Line 195: “values represent the group’s integrity in the school (of fish)” this needs justification

  • At the end of section 2.1, the description of IIT 3.0 terms used and their biological meanings has been added.

Fig. 3a legend title: write 0.05, instead of .05

  • The point has been fixed.

Line 226-228: Both (i) and (ii) should be rephrased. It is not clear at the moment.

  • The point has been fixed.

Fig. 4a: what is the meaning of the brown square at (0,0). That was supposed to represent N=4 and N=5 all timescales on top of each other? If there is no data in that range, then is this why it is defined as 0?

  • The graph just meant there is no value for N=4, 5. But it seems confusing. So, we removed them from the figure.

Line 245: Normalizing with the observed maximum value: this can have a crucial effect on the numbers, if a fully connected state occurs rarely. That would mean that an occasionally high value rescales the metric.

  • Most of the high Φ values associate with high occurrence of fully connected states. The only exception is "chasing" in N=2, but their high Φ value distribution also observed in high occurrence of fully connected states. Therefore, the normalization applied here works well throughout our analysis.

Fig. 6: the last bin of the x axis on all 4 panels states 3000. Is this correct? If so, still should be explained.

  • This is to show that a strong dependence of Φ on the average direction. The intermediate value is omitted due to space reasons. I added this description as a new caption in Fig 6.

Fig. 7: How does using only the top values affects this result?

  • The sample number does not affect the results; we also examined the other sample numbers (10 and 30) and produced the same results (Table S1).

Fig. 8: What n were used in the Pearson correlation tests? All data points? Are those data point statistically independent? Please explain/justify.

  • All data point (n=3200) for each time scale and added with some comment on it (Figure 8).

Discussion: I do not understand why the “original heap paradox” is discussed in details, rather its consequences/implications to group behaviour

  • We considerably rewrite the first half of the Discussion. On the other hand, the heap paradox was described as being the implication obtained from our results. To be sure, our study claims the possibility of a new classification using IIT 3.0, but there is no guarantee that that classification will be the same for other species. Other patterns could be observed in other species. What is important is what the causal analysis itself suggests, and we consider the heap paradox displays the core of the problem which IIT 3.0 evoke. The content of the above was also added to the Discussion.

Line 365-367: this is an example of the possible tautology raised in the major concerns

  • As we have already mentioned above, the new additions Figure 9 and Table 2 avoid the reviewer’s concern.

Table 2: the table legend is not very useful, as basically just repeats the title row of the table. Also what is the reason writing “(mm) per second”? Proper description here would help the reader

  • The point has been fixed.

Note:

I could not access the Supplementary Figures.

  • Since the link of Supplementary Figure didn’t work, they are attached as a separate data in the revised version.

We would like to sincerely appreciate your insightful and constructive comments and suggestions. We believe that these have greatly strengthened the paper.

Thank you once again for taking the time to review our manuscript.

Yours sincerely,

Takayuki Niizato

Department of Intelligent Interaction Technologies

University of Tsukuba

Tennodai 1-1-1, Tsukuba,

Ibaraki, Japan 305-8577

Reviewer 2 Report

This paper uses Integrated information theory (IIT) to analyze the collective behavior of a group of fish. The use of this technique to study a group of fish has been the subject of a previous publication. Here the authors add an analysis of the global parameters, the analysis of the group as a whole, and study the effect of time step on the results of the IIT. These new analyses provide new informations that are relevant for publication, especially the time step analysis that will certainly be interesting for group behavioural biologists.

Nevertheless, the current paper suffers from several problems that need to be improved:

- Some methodological informations are missing from the publication (I will detail them below) which makes the reading of some results difficult. I am aware that this has been described in detail in the publication [77] but it will be preferable to provide these informations here.

- The results of the study, and mainly those on the time step, are not highlighted in the discussion.

- A detailed correction of the English language and style is necessary.

On the content:

- The reasons for the activation or not of the fish is not explained for the local parameters. While this can be guessed for distance and field of view from Figure 1 (a), it is not very understandable for angular velocity. These must be absolutely detailed either in the text or in the legend of the figure.

- The authors state in the article that their definition of leadership differ from the classical literature. Nevertheless, at no point in the article is the method for detecting the presence of a leader in groups specified. As a consequence, the results presented in Figure 9 are difficult to read. In addition, the authors seem to explain that within the groups, the fish are either leaders, followers or neither. On the contrary, the presence of a leader should imply the existence of followers and vice versa. The authors must specify the method and the results obtained, as this is the basis for the definition of the groups they propose next (fission-fusion, leadership, boid-like).

On the form, in the reading order and in a non-exhaustive way :

- l.46 "elegantly" does not seem to fit the style of a scientific article.

- l.83 "TPM" is only defined on page 6. 

- l.144 Reference to section 2.2.1 whereas the text seems to refer to section 2.3. Also, a closing parenthesis is missing.

- l.180 Can you better define "MIP cut".

- l.210 "this is not true" Reformulate.

- Figure 3 (a) Can you add the legend of the shapes in the figure?

- l.238 In this part, could you add an explanation about the presence of low Ф(N) for groups of more than two fish?

- l.305 "the agents recognise can their environment" to be corrected.

- l.337 "Interestingly, this timescale approximately corresponds to the reaction time of the fishes [42]." Do the authors indicate that they found the reaction time of Plecoglossus altivelis ? In that case the result should be more explicit. If not, please indicate the reaction time of this species and the associated reference.

- l.338 "The graph shows that a school of five fishes at Dt = 20/120 s is because of the dominance of the local group integrity compared with that in other school sizes (Table 1)". I don't understand that sentence.

- l.360 "(as different systems from each other)" Explain this statement, since studies of group behavior, especially in fish, focus on small group sizes.

- l.365-370 This paragraph is redundant with the following paragraphs.

- Figure 9 The fundamental issues have been explained above. Nevertheless, the use of the same letters for different information ((L) for leader AND for local interaction) does not help the reading of this figure. Also in the captions, "time scale" is written in two words unlike the rest of the article.

- l.448 "How?" Please rephrase.

- l.402 The parenthesis is too long and breaks the reading of the sentence.

- l.405 I don't understand this sentence.

- Materials and Methods. Can you specify the duration of the experiments and the number of replicates for each group size?

Author Response

Responses to Reviewer 2:

Thank you again for taking the time to review our manuscript. We have addressed all of your comments, as described in the dialogue below.

Some methodological informations are missing from the publication (I will detail them below) which makes the reading of some results difficult. I am aware that this has been described in detail in the publication [77] but it will be preferable to provide these informations here.

  • We added a review of the previous research in the Introduction section. At the end of section 2.1, a description of the IIT 3.0 terms used in this paper and the correspondence between their biological meanings and the IIT terminology is given. We believe this seems to have made the outlook better than before.

The results of the study, and mainly those on the time step, are not highlighted in the discussion.

  • The first half of Discussion has been entirely revised.

A detailed correction of the English language and style is necessary.

  • The revised paper has been carefully reviewed by a native English speaker.

On the content:

The reasons for the activation or not of the fish is not explained for the local parameters. While this can be guessed for distance and field of view from Figure 1 (a), it is not very understandable for angular velocity. These must be absolutely detailed either in the text or in the legend of the figure.

  • The problem was that the picture of the fish was too big. To avoid such misunderstandings, we have redrawn the figures and added a detailed explanation of each example to the caption in Figure 2.

The authors state in the article that their definition of leadership differ from the classical literature. Nevertheless, at no point in the article is the method for detecting the presence of a leader in groups specified. As a consequence, the results presented in Figure 9 are difficult to read. In addition, the authors seem to explain that within the groups, the fish are either leaders, followers or neither. On the contrary, the presence of a leader should imply the existence of followers and vice versa. The authors must specify the method and the results obtained, as this is the basis for the definition of the groups they propose next (fission-fusion, leadership, boid-like).

  • The revised manuscript includes comparisons between the integrated information Φ and specific collective behaviors. The major difference between the revised version and the previous version is that the old version of Figure 9 was removed and the new version of Figure 9 and Table 2 were added. First of all, we changed the name "followership" to "chasing" in the classification. Since that leadership and followership are paired concepts with each other, this image invokes the reader’s confusions and misunderstandings. This is not what we aim for. For the same reason, we changed "Boid-like" to "interactive" because Boid already has a solid image. Our purpose of this paper is not to argue whether the systems of five-fish school is really BOID-like or not. For the rest of the items, the differences and similarities with previous terminology are clearly described in each item. The revised version shows that the four group sizes are neatly classified "on a numerical basis" and discusses their correspondence to actual group behavior. This allows for a much more analytical description than before the revision.

On the form, in the reading order and in a non-exhaustive way :

l.46 "elegantly" does not seem to fit the style of a scientific article.

  • The word “elegantly” was eliminated.

l.83 "TPM" is only defined on page 6.

  • The point has been fixed.

l.144 Reference to section 2.2.1 whereas the text seems to refer to section 2.3. Also, a closing parenthesis is missing.

  • The point has been fixed.

l.180 Can you better define "MIP cut".

  • We defined MIP cut at the end of section 2.1 and added its corresponding biological meanings.

l.210 "this is not true" Reformulate.

  • The point has been fixed.

Figure 3 (a) Can you add the legend of the shapes in the figure?

  • We added the shape caption on Figure 3 and 4 (a).

l.238 In this part, could you add an explanation about the presence of low Ф(N) for groups of more than two fish?

  • We rewrite this section.

l.305 "the agents recognise can their environment" to be corrected.

  • The point has been fixed.

l.337 "Interestingly, this timescale approximately corresponds to the reaction time of the fishes [42]." Do the authors indicate that they found the reaction time of Plecoglossus altivelis ? In that case the result should be more explicit. If not, please indicate the reaction time of this species and the associated reference.

  • We added the scientific name with reference.

l.338 "The graph shows that a school of five fishes at Dt = 20/120 s is because of the dominance of the local group integrity compared with that in other school sizes (Table 1)". I don't understand that sentence.

  • We revised this sentence.

- l.360 "(as different systems from each other)" Explain this statement, since studies of group behavior, especially in fish, focus on small group sizes.

  • This section has been entirely revised.

l.365-370 This paragraph is redundant with the following paragraphs.

  • This paragraph was entirely eliminated.

Figure 9 The fundamental issues have been explained above. Nevertheless, the use of the same letters for different information ((L) for leader AND for local interaction) does not help the reading of this figure. Also in the captions, "time scale" is written in two words unlike the rest of the article.

  • As we have noted earlier, we added the new Figure 9 and Table 2 with the detail analysis. In the revised manuscript, all classifications are based on the data analysis (Φ value, MIP cut and its causal structure).

l.448 "How?" Please rephrase.

  • We rewrite this sentence

  • We couldn’t find the correspondent sentence which the following comments suggested (the number of line seems reversed here). However the half of the Discussion section has been revised and the revised paper has been carefully reviewed by a native English speaker.

l.402 The parenthesis is too long and breaks the reading of the sentence.

l.405 I don't understand this sentence.

Materials and Methods. Can you specify the duration of the experiments and the number of replicates for each group size?

  • We listed all data on Table 3. In addition, a brief description of the experimental data (number of experiments and total time) was added at the beginning in Section 2.2.

We would like to sincerely appreciate your insightful and constructive comments and suggestions. We believe that these have greatly strengthened the paper.

Thank you once again for taking the time to review our manuscript.

Yours sincerely,

Takayuki Niizato

Department of Intelligent Interaction Technologies

University of Tsukuba

Tennodai 1-1-1, Tsukuba,

Ibaraki, Japan 305-8577

Reviewer 3 Report

In this manuscript, the authors analyze behavioral data from small groups of fish. The basic analysis uses Integrated Information Theory to measure the "integrity" (Phi) of fish schools varying from size 2 to size 5. The authors argue that their results produce a new classification that identifies the number of fish required to observe various aspects of collective behavior (leading and following, and "local" and "global" integrity).

At the level of describing the analysis that was performed, the manuscript appears fine (barring some minor issues). Yet at a conceptual level, for three major reasons listed below, the analysis appears to be fundamentally flawed. Given these fundamental issues and a number of other issues with clarity, it is difficult for me to envision how the manuscript might be modified into a publishable piece.

1) The data is being coarse-grained in a strange way that mixes and removes information about which behavior is being performed. The authors argue (in the paragraph starting on line 153) only that their coarse-graining using AND is better than one that uses OR. But why coarse-grain in this way to begin with? The IIT analysis is implicitly inferring a model that specifies Markovian probabilities of transition between various states of the system. It seems that the best way to build such a model would be to specify fish states in such a way that the state at time t could reasonably predict the state at time t + delta t. But the authors' setup mixes different behaviors in a way that makes it hard to interpret. A fish changing from an inactive to an active state could correspond to changing directions when it's already close to another fish, or getting closer to another fish when it's already changing directions. Why not keep these behaviors separate and measure probabilities with which fish, for instance, turn given closeness to others, get closer given that others are turning, etc.? It seems that this would create much more predictive and interpretable models. The authors' coarse-graining feels like an arbitrary shoehorning of the data into a form that can be used easily with IIT tools.

2) Even if these behaviors were kept separate, the setup would infer causal relationships only in uninterpretable ways because it neglects spatial structure (and so does not respect locality). For example: Say we find that state 1110 tends to follow state 1100. What does this mean? If fish 1 and 2 are active and fish 3 is not at time t, then fish 3 tends to become active at time t + delta t. This rule must hold regardless of whether fish 3 is far away from 1 and 2, between them, ahead of them, behind them, whether it can see them, etc. The only condition is that fish 3 is either not turning OR can't see any other fish OR is not close to any other fish. I find it nearly impossible to interpret such a model or believe that it might be accurate or predictive. (The predictiveness or quality of the model --- how well it describes the data --- is in fact never tested.)

3) The manuscript seems to implicitly assume that maximizing Phi with respect to parameters that define coarse-graining cutoffs provides information about the interactions that are driving collective behavior. For instance, line 384 states as a result that "the three-fish school prefers to use the complete visual field", using as evidence (if I'm reading Fig 3a correctly) that Phi is maximized when the visual field parameter is not restricted (so that individuals can be counted as active regardless of where neighbors appear). Interpreting this as evidence that fish in the school are actually using the full visual field seems like a large stretch to me, and would need to be explained in more detail. Is varying parameters to maximize Phi a legitimate way to infer their true values?

Besides these fundamental issues, much of the interpretation remains vague and undersupported.  Large points throughout the manuscript remain very unclear to me. For instance, the discussion makes a large effort to explain the difference between "external" and "internal" analysis, but the distinction remains nebulous. The external approach is described as one that "only represents the predictability of each pair of individuals from their external behaviours" (line 418). How is the IIT approach any different? Isn't it using the same observed behavioral data to reach its conclusions? What's different about behavior being "internal" as opposed to "external"? Line 420 states that "the internal perspective provides us with different perspectives because the collective behaviour from external perspectives is needless". Why is it needless? What is the "external perspective"?

The measure plotted in Figure 7 is interpreted as measuring whether "local" or "global" interactions dominate the behavior, with the interpretation spelled out in the paragraph starting on line 326. I see no evidence presented that this interpretation is valid. For example, line 330 states "The local group integrity tends to be lower in short time scales because the noisy movements of the fish weaken their predictabilities." Where is the evidence for this connection? Besides this, I am not convinced that it makes sense to compare Phi values calculated in two completely different ways (one using local parameters and one using global parameters). And why use the top 20 values of Phi in each case (see line 312)? The measure seems very arbitrary.

In many cases, the structure of sentences makes them very difficult to parse. For instance, line 452 states "The computation of these average values implicitly contains the decision group’s criterion of the observer because we assume the given data set already forms the group." This sentence makes little sense as it stands, but after unpacking the rest of the paragraph, perhaps the authors mean to say "The computation of these average values implicitly assumes that an observer has decided to consider the set of birds as a group"? (And even if this is what the authors meant, I find it confusing to say that scale-free correlations can only be interpreted in terms of aggregated group behavior. Correlations between distant birds can be computed whether or not we include the other birds in the calculation.) There are numerous such examples of awkward grammatical construction throughout the text that seriously impair readability. (For instance: line 192 "Considering that the past movements to the network may fail to determine the pure inter-relations among the fishes" --- What are "past movements to the network"? What are "pure inter-relations"?)

The manuscript refers to plots displaying Phi as a function of cutoff parameters as "distributions", saying that they are "concentrated" in various regions (e.g. Figure 3a caption). This seems to be confusing the function Phi with a probability distribution.

Table 1 appears to be measuring the significance of the difference between (Phi_global - Phi_local) across pairs of group sizes. In what sense can this difference be considered statistically significant? Where is there a distribution over Phi values? Are you estimating uncertainties in Phi measured from the data? (That doesn't appear to be the case.)

Author Response

Responses to Reviewer 3:

Thank you again for taking the time to review our manuscript. We have addressed all of your comments, as described in the dialogue below.

The data is being coarse-grained in a strange way that mixes and removes information about which behavior is being performed. The authors argue (in the paragraph starting on line 153) only that their coarse-graining using AND is better than one that uses OR. But why coarse-grain in this way to begin with? The IIT analysis is implicitly inferring a model that specifies Markovian probabilities of transition between various states of the system. It seems that the best way to build such a model would be to specify fish states in such a way that the state at time t could reasonably predict the state at time t + delta t. But the authors' setup mixes different behaviors in a way that makes it hard to interpret. A fish changing from an inactive to an active state could correspond to changing directions when it's already close to another fish, or getting closer to another fish when it's already changing directions. Why not keep these behaviors separate and measure probabilities with which fish, for instance, turn given closeness to others, get closer given that others are turning, etc.? It seems that this would create much more predictive and interpretable models. The authors' coarse-graining feels like an arbitrary shoehorning of the data into a form that can be used easily with IIT tools.

  • The coarse-graining is a constraint of IIT 3.0 itself, which can only handle discrete quantities; IIT can indeed handle continuous quantities, but it is available only IIT 1.0 and 2.0. Although IIT 3.0 can only handle discrete systems, the discrete system enable us to consider Φ for each state (e.g. 00, 01, 01, 11). This discreteness enable us to compare the IIT 3.0 analysis with the actual collective behaviour.
  • The coarse-grainedness we have presented has a biological meanings. The combination of AND operators corresponds to the BOID model which is the most well-applicable model to the animal collective behaviours. From this combination, we can obtain many parameter combination including visual field only; however, generally, the these separate parameters tend to decrease the Φ values.
  • The BOID-like interaction was used for two reasons. First, this is for biological reasons. The other is so that we can immediately see in the model whether our analysis is just an artificial result. A comparison on this model is detailed in Niizato 2020. For a review of this paper, I have added them to Introduction section.

Even if these behaviors were kept separate, the setup would infer causal relationships only in uninterpretable ways because it neglects spatial structure (and so does not respect locality). For example: Say we find that state 1110 tends to follow state 1100. What does this mean? If fish 1 and 2 are active and fish 3 is not at time t, then fish 3 tends to become active at time t + delta t. This rule must hold regardless of whether fish 3 is far away from 1 and 2, between them, ahead of them, behind them, whether it can see them, etc. The only condition is that fish 3 is either not turning OR can't see any other fish OR is not close to any other fish. I find it nearly impossible to interpret such a model or believe that it might be accurate or predictive. (The predictiveness or quality of the model --- how well it describes the data --- is in fact never tested.)

  • By adding a new Figure 9, Table 2, we discuss in detail how IIT 3.0 can be associated with actual collective behaviors. We hope that this will deepen the reviewer's interpretation. The list of collective states that appear in the parameter region with a high mean Φ narrows. As shown in the newly added Figure 9, the two states of all ON states and single OFF collective states account for more than 99% of the whole process. For them, it is sufficiently possible to relate them to individual behavior in real data.

The manuscript seems to implicitly assume that maximizing Phi with respect to parameters that define coarse-graining cutoffs provides information about the interactions that are driving collective behavior. For instance, line 384 states as a result that "the three-fish school prefers to use the complete visual field", using as evidence (if I'm reading Fig 3a correctly) that Phi is maximized when the visual field parameter is not restricted (so that individuals can be counted as active regardless of where neighbors appear). Interpreting this as evidence that fish in the school are actually using the full visual field seems like a large stretch to me, and would need to be explained in more detail. Is varying parameters to maximize Phi a legitimate way to infer their true values?

  • As stated in Section 2.2, I am not stating that the visual parameter, for example, is this actual perception, but rather an effective perception. The actual perception and effective perception range are also strictly distinguished in the field of collective behaviour. Many analyses use a (complete) visual field of the VICSEK type interaction, but they do not claim to perceive its extent. (I mentioned this in Section 2.2)
  • Setting parameters to the largest Φ: Φ values are often associated with critical phenomena, etc., and seem reasonable as far as their context is concerned. Furthermore, the analysis of maximal values for certain parameters can be found in other studies besides this one, such as the analysis of transfer entropy (see Introduction).

Besides these fundamental issues, much of the interpretation remains vague and undersupported. Large points throughout the manuscript remain very unclear to me. For instance, the discussion makes a large effort to explain the difference between "external" and "internal" analysis, but the distinction remains nebulous. The external approach is described as one that "only represents the predictability of each pair of individuals from their external behaviours" (line 418). How is the IIT approach any different? Isn't it using the same observed behavioral data to reach its conclusions? What's different about behavior being "internal" as opposed to "external"? Line 420 states that "the internal perspective provides us with different perspectives because the collective behaviour from external perspectives is needless". Why is it needless? What is the "external perspective"?

  • The external perspective focuses on the actual behaviour of their objective, while the internal perspective focuses on the causal structure of the extracted network without looking directly at the behaviour. The corresponding behaviours are the actualization from their potential causal structure. This difference shows a sharp contrast when considering information process inside the group. For instance, when we define the "leadership" in the traditional meaning, we measure the information flow between two individuals and decide which flow is dominant (e.g. transfer entropy). On the other hand, the "leadership" that we used here is not a pair of the individuals but the pair of sub-groups induced by MIP cut. As we saw in the newly added Fig. 9, this division of subgroups is not necessarily correlated with positional relationships (three-fish school). From an internal perspective, this division is the result of the relationship between subsystems from whole system transitions. In this study, the correlation between external behaviour (leadership) and internal process (leadership induced by MIP cut) makes it difficult to distinguish between the external and internal perspectives. Our additional analysis may have cleared that misunderstanding.
  • “Needless” was too strong expression, so I removed it.

The measure plotted in Figure 7 is interpreted as measuring whether "local" or "global" interactions dominate the behavior, with the interpretation spelled out in the paragraph starting on line 326. I see no evidence presented that this interpretation is valid. For example, line 330 states "The local group integrity tends to be lower in short time scales because the noisy movements of the fish weaken their predictabilities." Where is the evidence for this connection? Besides this, I am not convinced that it makes sense to compare Phi values calculated in two completely different ways (one using local parameters and one using global parameters). And why use the top 20 values of Phi in each case (see line 312)? The measure seems very arbitrary.

  • "The local group integrity tends to be lower in short time scales because the noisy movements of the fish weaken their predictabilities." This sentence comes from a general assumption of the IIT 3.0 characteristics, but it is difficult to demonstrate this in the pyphi-framework. However, the descriptions (L326-L335) have been removed since they evoke the reader's confusion.
  • We believe that the meaning of Top-down and Bottom-up become more clear for adding analysis of the meaning of local interaction of five-fish schools in newly added Figure 9.
  • The sample number does not affect the results; we also examined the other sample numbers (10 and 30) and produced the same results (Table S1).

In many cases, the structure of sentences makes them very difficult to parse. For instance, line 452 states "The computation of these average values implicitly contains the decision group’s criterion of the observer because we assume the given data set already forms the group." This sentence makes little sense as it stands, but after unpacking the rest of the paragraph, perhaps the authors mean to say "The computation of these average values implicitly assumes that an observer has decided to consider the set of birds as a group"? (And even if this is what the authors meant, I find it confusing to say that scale-free correlations can only be interpreted in terms of aggregated group behavior. Correlations between distant birds can be computed whether or not we include the other birds in the calculation.) There are numerous such examples of awkward grammatical construction throughout the text that seriously impair readability. (For instance: line 192 "Considering that the past movements to the network may fail to determine the pure inter-relations among the fishes" --- What are "past movements to the network"? What are "pure inter-relations"?)

  • The revised paper has been carefully reviewed by a native English speaker.
  • "The computation of these average values implicitly assumes that an observer has decided to consider the set of birds as a group”? The reviewer’s interpretation is right. Of course, as the reviewer says, it is possible to calculate them in the way of reviewer's suggestions. However, collective behaviour analysis begins by treating the observed objects as if they form one collective. The scale-free correlation also shares this premise since their discussion information likens the transfer in the flock to the brain. We added this abstract in Discussion.

The manuscript refers to plots displaying Phi as a function of cutoff parameters as "distributions", saying that they are "concentrated" in various regions (e.g. Figure 3a caption). This seems to be confusing the function Phi with a probability distribution.

  • We removed statements that could be misinterpreted as such a probability distribution.

Table 1 appears to be measuring the significance of the difference between (Phi_global - Phi_local) across pairs of group sizes. In what sense can this difference be considered statistically significant? Where is there a distribution over Phi values? Are you estimating uncertainties in Phi measured from the data? (That doesn't appear to be the case.)

  • The difference between the data of all combinations of the values of each individual for Delta_t=20/120 is calculated, which is written in the caption of Table 1.

We would like to sincerely appreciate your insightful and constructive comments and suggestions. We believe that these have greatly strengthened the paper.

Thank you once again for taking the time to review our manuscript.

Yours sincerely,

Takayuki Niizato

Department of Intelligent Interaction Technologies

University of Tsukuba

Tennodai 1-1-1, Tsukuba,

Ibaraki, Japan 305-8577
